# Immune responses to *Mycobacterium tuberculosis* membrane-associated antigens including alpha crystallin can potentially discriminate between latent infection and active tuberculosis disease

Shashi Kant Kumar[1], Suvrat Arya[1], Amita Aggarwal[1], Prerna Kapoor[2], Alok Nath[3], Ramnath Misra[1], Sudhir Sinha[1]*

**1** Department of Clinical Immunology & Rheumatology, Sanjay Gandhi Postgraduate Institute of Medical Sciences, Lucknow, India, **2** DOTS Centre, Sanjay Gandhi Postgraduate Institute of Medical Sciences, Lucknow, India, **3** Department of Pulmonary Medicine, Sanjay Gandhi Postgraduate Institute of Medical Sciences, Lucknow, India

* ssinha@sgpgi.ac.in, sinha.sudhir@gmail.com

## Abstract

Changes in expression of membrane antigens may accompany the transition of *Mycobacterium tuberculosis* (Mtb) from 'dormant' to 'active' states. We have determined whether antibody and T cell responses to Mtb membrane (MtM)-associated antigens, especially the latency-induced protein alpha crystallin (Acr), can discriminate between latent tuberculosis infection (LTBI) and active TB (ATB) disease. Study subjects comprised a previously described cohort of healthcare workers (HCWs, n = 43) and smear-positive ATB patients (n = 10). HCWs were further categorized as occupational contacts (OC, n = 30), household contacts of TB (HC, n = 8) and cured TB (CTB, n = 5). Levels (ΔOD) of serum antibody isotypes (IgG, IgA and IgM) were determined by ELISA and blood T cell proliferative responses were determined by flow cytometry using Ki67 protein as marker for DNA synthesis. Antibodies to MtM and Acr were predominantly IgG and their levels in HCWs and ATB did not differ significantly. However, HCWs showed a significantly higher level of anti-MtM IgM and a significantly lower level of anti-Acr IgA antibodies than the ATB patients. Also, a larger proportion of HCWs showed a high (>1) $\Delta OD_{Acr}/\Delta OD_{MtM}$ ratio for IgG. HCWs also showed a higher, though not significantly different from ATB, avidity of anti-MtM (IgG) antibodies. A higher proportion of HCWs (35% of OC, 62.5% of HC and 20% of CTB), compared with ATB (10%) showed a positive T cell response to Acr along with significant difference (P <0.05) between HC and ATB. A significant correlation (r = 0.60, P <0.0001) was noted between T cell responses of HCWs towards Acr and MtM (reported earlier by us) and both responses tended to decline with rising exposure to the infection. Even so, positive responses to Acr (38.5%) were significantly lower than to MtM (92%). Neither antibody nor T cell responses to either antigen appeared affected by BCG vaccination or reactivity to tuberculin. Results of the study suggest that the levels of IgM antibodies to MtM, IgA antibodies to Acr and

**Data Availability Statement:** All relevant data are within the paper and its Supporting Information files.

**Funding:** The authors received no specific funding for this work.

**Competing interests:** The authors have declared that no competing interests exist.

proliferative T cell responses to both the antigens can potentially discriminate between LTBI and active TB disease. They also underscore the necessity of SOPs for antibody assays.

## Introduction

A sustained 20% decline in incidence of tuberculosis (TB) is needed to meet the goals of 'End TB' strategy whereas current rate of decline is only about 2% [1]. The situation is particularly alarming in TB hyperendemic countries such as India, where a significant proportion of disease remains undetected [1]. A major population of these countries also harbors 'latent TB infection (LTBI)' defined as 'a state of persistent immune responsiveness to *Mycobacterium tuberculosis* (Mtb) without clinically manifested disease'. The lifetime (mostly 5-year) risk of reactivation of LTBI leading to active TB disease is about 10% [1]. In this scenario, a biomarker which can distinguish between quiescent and active infections may help identify persons who could benefit from prophylactic measures [2].

The widely used tuberculin skin test (TST) and interferon-gamma release assays (IGRAs) estimate an existing immune response to Mtb hence provide only a 'presumptive' evidence of infection. Given that they show low specificity and sensitivity in low- and middle-income countries, WHO strongly recommends that both (IGRAs and TST) should not be used for diagnosis of TB or identification of persons at risk of developing TB [3]. Further, due to their comparable performances in different settings, replacing the TST by IGRAs as a public health intervention is not recommended [3]. The fact that neither test has the desired sensitivity to detect LTBI is also evident from a recent study from north India in which, during the follow-up period, incidence of TB in test-positive (TST or IGRA) and test-negative contacts was comparable [4]. Another recent report has suggested that certain differences in adaptive immunity in a subset of persons who are exposed to Mtb could be responsible for their test-negativity for TST and IGRA [5]. The authors did however not rule out the possibility that such test-negative subjects could actually be harboring LTBI.

Constituents of mycobacterial plasma membrane are potent inducers of innate and adaptive immune responses in humans [6–8]. We have reported on their capability to induce T cells of Indian leprosy and TB patients [9] as well as healthcare workers (HCWs) [10]. A genome-wide search has identified over 80 Mtb proteins, mostly of membrane origin, which were strong inducers of T cells from subjects with LTBI [11]. Recently, Li et al [12] used a panel of Mtb membrane proteins to identify antigens which produced stronger T cell responses in TB patients than did ESAT6 (a component of IGRA). With respect to B cell responses also, we have shown that sera of healthy Indians contain significantly higher levels of antibodies to membrane-associated than cytosolic antigens of Mtb [13, 14]. In another study, serum antibodies of TB patients reacted strongly with Mtb membrane (MtM) proteins [15]. In yet another study, antibodies secreted by B cells of TB patients and HCWs preferentially targeted MtM [16]. In an analysis of sera of TB suspects it was concluded that the Mtb 'immunoproteome' is rich in membrane-associated or secreted proteins and not cytoplasmic proteins [7]. One study [17] has reported that active TB patients produce low avidity antibodies, along with low IgG/IgM ratio, to 'cell surface' antigens of Mtb.

The Mtb membrane-associated heat-shock protein alpha crystallin (Acr/Rv2031c/HspX) is believed to sustain the bacilli during latent or dormant phase of infection [18, 19]. Its abundant production by dormant bacilli and restriction to Mtb complex [19, 20] qualify Acr as a potential biomarker for LTBI. Indeed, it is a strong inducer of both T and B cell responses and

figures among most immunogenic proteins of Mtb [7, 21]. Compared to active TB disease, persons with LTBI have shown a stronger T cell response to Acr [21–23] which was not influenced by BCG vaccination [23]. High levels of anti-Acr antibodies have been reported in TB patients and their contacts [21, 24, 25], HCWs exposed to Mtb [26] and cerebrospinal fluid of patients with tuberculous meningitis [27]. Conversely, some studies have found lower antibody levels in active than in inactive or latent TB [28]. Some of the studies have laid emphasis on isotype of anti-Acr antibody. Two of them have reported that IgA and IgG levels are higher in patients than in contacts or healthy subjects [29, 30] and one study has reported that children with TB have higher IgG and IgM levels as compared to healthy children [31]. Another study has found that the level of anti-Acr IgM is significantly higher in LTBI than in active TB [32].

The above-mentioned studies suggest that antibody and T cell responses to Mtb membrane and the membrane-associated antigen Acr can potentially discriminate between LTBI and active TB disease. It is therefore eminently desirable to explore if these responses can be translated into the much needed biomarkers for latent and/or active infections, particularly in TB hyperendemic situations. A critical requirement for their fair evaluation, which has not been met in past studies, is that they should be determined in the same study population. It also appears pertinent to ascertain whether antibody isotypes have any role as predictors of the state of infection. We have tried to address these and related queries in this study using a cohort of Indian HCWs and TB patients whose baseline characteristics, along with T cell responses to PPD (purified protein derivative of Mtb) and MtM were reported previously by us [10]. Our choice for HCWs was based on the fact that they, compared to population-at-large, carry a greater risk of contracting the Mtb infection [1, 33].

## Materials and methods

### Ethics statement

The study protocol was approved by the Institutional Ethics Committee of Sanjay Gandhi Postgraduate Institute of Medical Sciences (SGPGIMS), Lucknow (IEC Code No. 2016-149-IMP-EXP). Written informed consent was obtained from all participants.

### Materials

A list of purchased materials used in this study is annexed as S1 Text.

### Study subjects

The study subjects (n = 53) comprised a previously described cohort of HCWs and TB patients [10]. In brief, 30 of the 43 HCWs who had not lived with a TB patient in their households were classified as occupational contacts (OC). Eight HCWs, who had lived (> 1 year prior to the study) with a bacteriologically positive patient for at least 3 months were classified as household contacts (HC, according to WHO/HTM/TB/2012.9). Five HCWs had a history of cured TB (CTB, according to WHO/HTM/TB/2013.2). BCG scar was present in 81% and positive TST (≥ 10 mm induration) was recorded in 56% HCWs [10]. All 10 TB patients had bacteriologically confirmed active disease (according to WHO/HTM/TB/2013.2), with sputum-smears positive (1+ to 3+) for acid-fast bacilli. Xpert MTB/RIF assay (WHO/HTM/TB/2013.16) was performed in 2 patients who were suspected of drug resistance. However, both were found sensitive for RIF. Clinical records of remaining 8 patients did not suggest drug resistance. Nine patients were tested and found seronegative for HIV. In the remaining one patient (transferred to us from another clinic) also, we did not suspect HIV from her medical history.

## Blood samples

Blood was collected by standard venipuncture in sodium heparin tubes (1 ml) for T cell assays and in plain tubes (3 ml) to prepare sera for antibody assays.

## Mtb antigens

Mtb cell membrane (MtM) was isolated using the previously reported protocol [10]. In brief, 3 week old culture of Mtb (strain H37Ra, ATCC 25177) on Lowenstein-Jensen (L-J) medium was harvested and bacterial sediment was washed and suspended in PBS (0.2 g wet wt/ ml). The cell lysate obtained by sonication was centrifuged (23,000g) to settle unbroken cells and cell wall debris. The supernatant was re-centrifuged (150,000g) to obtain the membrane (sediment) and cytosol (supernatant). Membrane was washed and reconstituted with PBS and protein was estimated using a modified Lowry's method suitable for membrane proteins [34]. This method was converted to micro-well plate format as follows. Dilutions of test protein and bovine serum albumin (BSA, as standard) were dispensed in flat-bottom 96-well plates (100 μl/ well). To each well, 200 μl alkaline copper reagent (containing 1% SDS) was added and mixed immediately by repeated pipeting. After 15 min, 20 μl of 1 N Folin's reagent was also added to each well and mixed immediately. Color was developed (30 min) and absorbance was read at 750 nm on a plate reader. Standard curve was prepared by plotting the average (blank corrected) OD values for the applied BSA concentrations. The recombinant Mtb Acr protein was purchased from Lionex GmbH (Braunschweig, Germany, Cat. No. LRP-0019C). All protein aliquots were stored at -80˚C. As a quality control measure, identity of Mtb growing on the L-J medium was validated periodically by immunochromatographic detection of MPT64 antigen [35] using a kit ('SD Bioline TB Ag MPT64 Rapid', Abbott, USA).

## Enzyme-linked immunosorbent assay

A previously described protocol [13, 14] was used after necessary modifications. The used antigen concentrations and serum dilutions had provided optimal results during protocol development. Antigens (10 μg/ml MtM or 1 μg/ml Acr) in coating-buffer (0·05 M carbonate, pH 9·5) or coating-buffer alone were dispensed (50 μl/well) in U-bottom ELISA plates (Nunc Maxisorp) and incubated overnight at 4˚C. After washing with tris-buffered saline (0·05 M Tris, 0·1 M NaCl, pH 7·4) containing 0·05% Tween 20 (TBS-T), the plates were incubated (2 h, 37˚C) with blocking solution (2% skimmed milk powder dissolved in TBS-T, 100 μl/well). After removing the blocking solution, test sera (diluted 1:500 for MtM and 1:100 for Acr, in 1% milk-TBS-T) were dispensed in antigen- and buffer-coated wells (50 μl/ well, in duplicate wells) and incubated (2 h, 37˚C). Plates were later washed with TBS-T and incubated (50 μl/ well, 2 h, 37˚C) with affinity-purified peroxidase-conjugated antibodies to human IgG (diluted 1:4000 in 1% milk-TBS-T), IgA and IgM (both diluted 1:2000). Plates were finally washed with TBS-T and the substrate solution (0·04% o-phenylene diamine + 0·03% $H_2O_2$ in 0·05 M citrate phosphate buffer, pH 5) was added (50 μl/ well) and incubated at room temperature (20 min, in dark). Reaction was stopped by adding 7% $H_2SO_4$ (50 μl/ well). ODs were read at 492 nm on a plate reader. For each serum, difference in mean ODs of antigen- and buffer-coated wells was calculated and expressed as ΔOD.

## Determination of subclass of IgG antibodies

A modification of the ELISA protocol described above was used. Briefly, after incubation of antigen-coated wells with test sera and subsequent washings, monoclonal antibodies to human IgG1, G2, G3 or G4 were added (50 μl/ well) and incubated (2 h, 37˚C). After another round of

washing, wells were incubated (2 h, 37˚C) with peroxidase-conjugated anti-mouse IgG antibody (diluted 1:1000). Washed plates were finally incubated with substrate and stop solutions, and ODs were read.

### Determination of avidity index

A previously reported protocol [36] was used for estimation of avidity index (AI) of IgG antibodies to MtM. In brief, the ELISA protocol described above was followed up to the stage of incubation of antigen-coated wells with test sera. Later, washed wells (in duplicate) were filled (50 μl/ well) with either PBS-T (PBS containing 0.05% Tween 20) or PBS-T containing 8 M urea and incubated for 10 min at 37˚C. Washed wells were treated with peroxidase-conjugated secondary antibody, substrate and stop solutions as per above ELISA protocol. AI was calculated as follows: mean OD of urea-treated wells/ mean OD of untreated wells x 100.

We also tried an alternative protocol [5] for avidity determination. Accordingly, ELISA wells (after incubation with test sera) were treated with increasing concentrations of urea (0, 2, 4, 6 and 8 M) in order to determine its IC50 (50% inhibitory concentration). However, in 5 out of 6 tested sera, IC50 values were not achieved (S1 Fig); the reason for which could be difference in the nature of test antigens (PPD vs. MtM) and/or avidity of corresponding antibodies.

### T cell proliferation assay

We used the previously described protocol [10] with some modifications. In brief, blood samples (diluted 1:10 in RPMI medium) were dispensed in 24-well culture plates (1 ml/ well) and incubated with test antigen (Acr, 5 μg/ml) and controls (culture medium as negative and PHA, 5 μg/ml, as positive control) for 5 days in a $CO_2$ incubator. The cells were treated with EDTA (2 mM) and harvested. To re-suspended cells, fluorescent anti-CD3 antibody was added and incubated. For RBC lysis, cells were treated (2 ml/tube, 15 min, in dark) with a lysing solution (150 mM ammonium chloride, 14 mM sodium bicarbonate, 0.1 mM EDTA and 1% paraformaldehyde). Leukocytes (sediment) obtained after centrifugation were washed with PBS and fixed with 2% para-formaldehyde. Cells were again washed with PBS containing 0.05% BSA (PBS-BSA) and permeabilized with 0.2% Triton X100. After another wash with PBS-BSA, cells were incubated with fluorescent anti-Ki67 antibody. Cells were finally washed with PBS-BSA and re-suspended in PBS. Data on $10^5$ cells in 'lymphocyte gate' was acquired on a FACS Canto-II flow cytometer (BD) and analysed using FlowJo software v10.0.7 (Tree Star, USA). The gating strategy is shown in S2 Fig. Cut-off for a positive response was determined as 0.7% (= mean + 3 x SD of all responses to medium alone).

We also looked for a possible loss in T cell viability or their capacity to proliferate after 5 days in culture. Whole blood cultures were harvested and stained for CD3 as described above. After washing with PBS, cells were resuspended in PBS (0.5 ml). Without lysing the RBCs, 10 μl (1 μg) propidium iodide (PI) was added to each sample tube and immediately thereafter (within 5–10 min) cells were acquired on a flow cytometer [37]. For positive control, cells were fixed with 2% paraformaldehyde after CD3 staining and permeabilized with 0.2% Triton X-100 prior to PI treatment. S3 Fig shows results for a smear-positive (3+) ATB patient. In this case, over 90% of T cells remained viable after 5 days in culture and also showed a strong (95%) proliferative response to PHA.

### Statistical analysis

As most datasets did not show normal distribution, they were analysed by non-parametric methods. Differences between grouped datasets were computed by Kruskal-Wallis test and

those between two datasets were computed by either Wilcoxon matched pair signed rank test (for paired data) or Mann-Whitney test (for unpaired data). Correlation between two datasets was computed by Spearman's method, and Fisher's exact test (using 2x2 contingency tables) was used to compare two proportions. P values (two-tailed) of $< 0.05$ were considered as significant. All statistical analyses were performed using GraphPad Prism software.

## Results

### Study population

Demography of the study cohort (n = 53), comprising 43 HCWs and 10 ATB patients, has been described in our previous report [10]. Key demographic data are also provided in Materials and Methods.

### Anti-MtM IgM and anti-Acr IgA antibodies can potentially discriminate between LTBI and active TB

Isotype-specific antibody responses of the study subjects against Mtb antigens- MtM and Acr are shown in Fig 1 and Table 1. The most readily detectable antibody isotype against both antigens was IgG. Anti-MtM IgG levels in OC, HC and ATB were significantly higher than the corresponding IgA and IgM levels (Fig 1A, S1 Table). Similarly, anti-Acr IgG levels were significantly higher than IgA levels in OC and HC and IgM levels in OC, HC and CTB (Fig 1B,

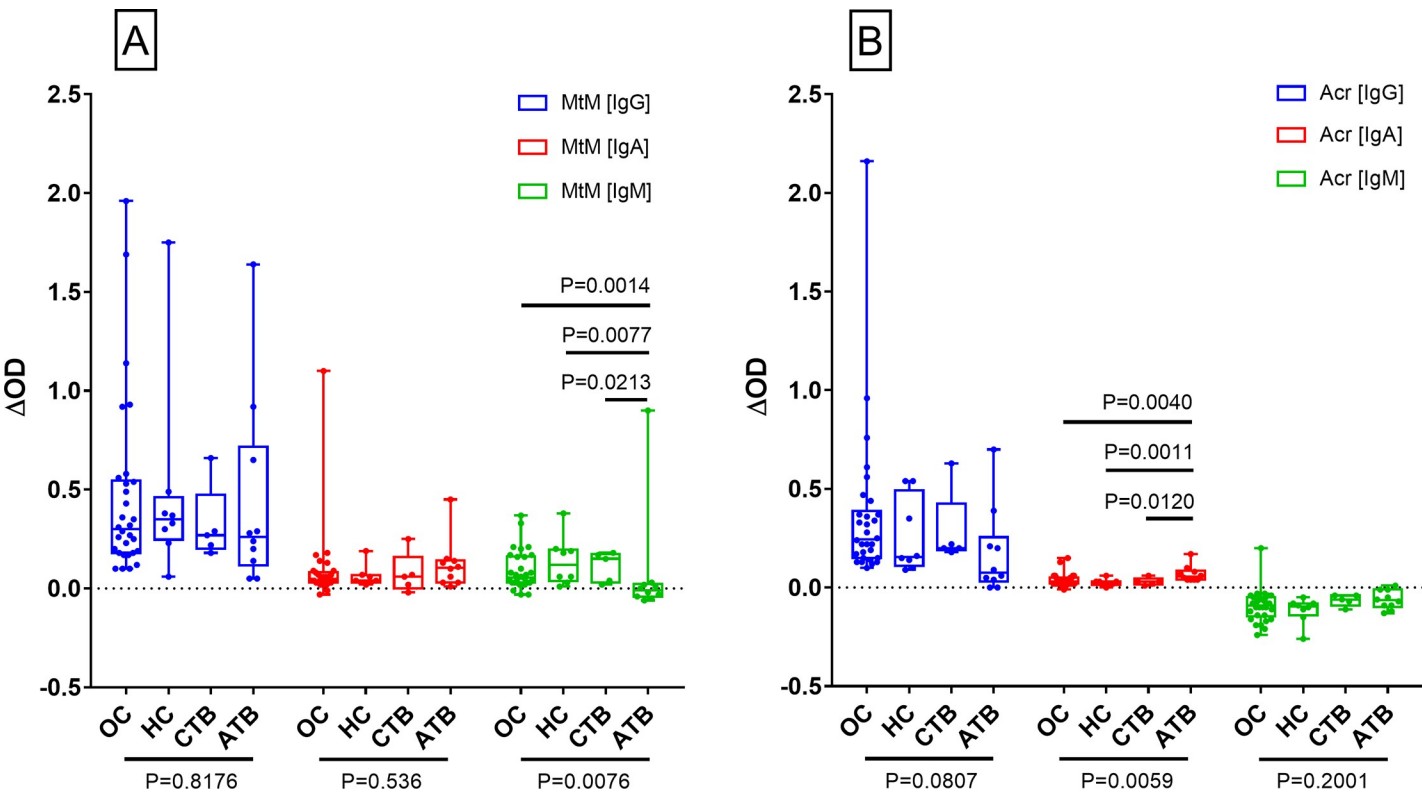

**Fig 1.** Levels (ΔOD) of antibody isotypes (IgG, IgA and IgM) against MtM (panel A) and Acr (panel B) in HCWs (OC, n = 30; HC, n = 8; and CTB, n = 5) and ATB patients (n = 10). Each column shows individual values (dots) along with median and IQR (boxes). Whiskers connect minimum and maximum values. P values ($< 0.05$ considered as significant) for multiple comparisons (Kruskal-Wallis test) for each Ig class across 4 groups and 2 antigens are shown below x axis (below the horizontal lines). P values for differences within the groups (Mann-Whitney test) which had shown significant differences in Kruskal-Wallis test are given on top of corresponding columns.

**Table 1. Levels of antibody isotypes against MtM and Acr in study subjects.**

|  | MtM | | | Acr | | |
|---|---|---|---|---|---|---|
|  | IgG | IgA | IgM | IgG | IgA | IgM |
| OC@ | 0.30 (0.18–0.55)# | 0.04 (0.03–0.08) | 0.06 (0.03–0.16) | 0.25 0.15–0.39) | 0.03 (0.02–0.05) | -0.09 (-0.15- -0.05) |
| HC | 0.35 (0.25–0.46) | 0.04 (0.03–0.07) | 0.12 (0.04–0.20) | 0.16 (0.11–0.49) | 0.03 (0.01–0.03) | -0.10 (-0.14- -0.08) |
| CTB | 0.27 (0.20–0.48) | 0.06 (0–0.16) | 0.15 (0.03–0.18) | 0.20 (0.19–0.43) | 0.03 (0.02–0.05) | -0.06 (-0.09- -0.04) |
| ATB | 0.26 (0.12–0.72) | 0.11 (0.03–0.14) | -0.01 (-0.04–0.02) | 0.08 (0.03–0.26) | 0.06 (0.04–0.09) | -0.07 (-0.10- -0.01) |

@ Subject category (OC, occupational contact; HC, household contact; CTB, cured TB; ATB, active smear-positive TB).

# Median ΔOD (inter-quartile range, IQR).

S1 Table). Regarding other 2 antibody isotypes, levels of anti-MtM IgA and IgM did not differ in any of the 4 categories. However, anti-Acr IgA levels were significantly higher than IgM in OC, HC and ATB (S1 Table). Conspicuously, almost all ΔOD values for anti-Acr IgM were < 0, which occurred because OD values of buffer-coated ELISA wells were higher than those of antigen-coated wells. This phenomenon is further described below.

Though the four subject categories did not differ from each other with respect to IgG responses against MtM or Acr (Fig 1), some differences were seen in corresponding IgM and IgA responses. The anti-MtM IgM antibody levels were significantly higher and anti-Acr IgA levels were significantly lower in HCWs (OC/HC/CTB) than in ATB. ROC (receiver operating characteristic) curves (Fig 2) also indicated that levels of both the antibodies could potentially discriminate between HCWs and ATB. AUC (ATB vs HCW pool) for anti-MtM IgM was 0.84 (95% CI, 0.65–1.02, P = 0.0009) and that for anti-Acr IgA, 0.83 (95% CI, 0.72–0.94, P = 0.0012). The ROC curves for all 3 HCW categories are also depicted in Fig 2. Accordingly,

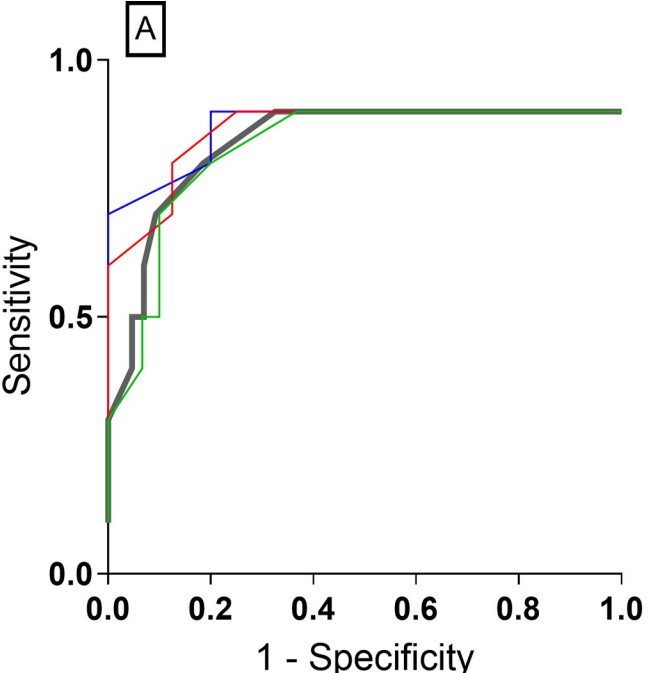
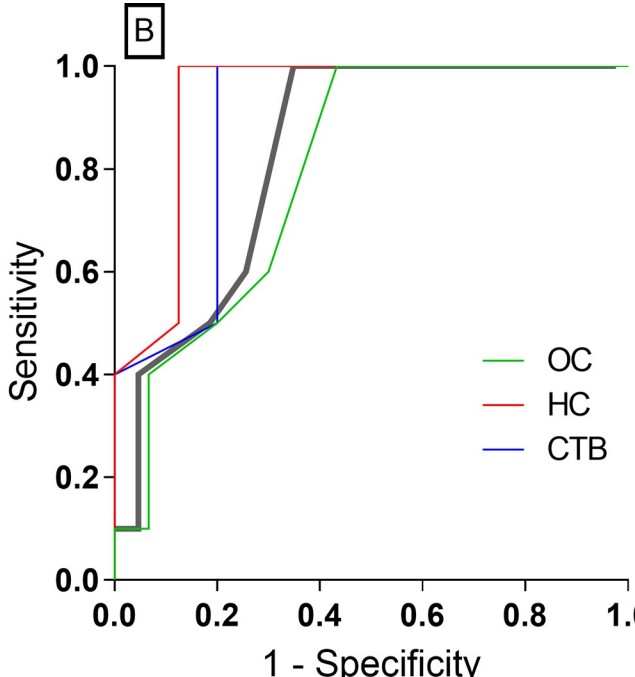

**Fig 2.** ROC curves with anti-MtM IgM (panel A) and anti-Acr IgA (panel B) antibody levels in HCWs (n = 43) and active TB (n = 10). The thick grey curve in both panels shows results with entire HCW pool and thin, colored curves (key in panel B) depict results for each HCW category: OC (n = 30), HC (n = 8) and CTB (n = 5).

AUC for anti-MtM IgM were as follows: OC, 0.83 (95% CI, 0.64–1.02, P = 0.0022); HC, 0.86 (95% CI, 0.67–1.06, P = 0.01) and CTB, 0.87 (95% CI, 0.67–1.07, P = 0.0235). Likewise, AUC for anti-Acr IgA were: OC, 0.80 (95% CI, 0.65–0.93, P = 0.0057); HC, 0.93 (95% CI, 0.80–1.07, P = 0.0022) and CTB, 0.89 (95% CI, 0.68–1.1, P = 0.0169). These data also suggest that the AUC values for HCW pool and HCW categories were comparable.

We also looked for IgG subtype(s) against MtM in a subset of sera (n = 12, S4 Fig). IgG2 was found to be the most readily detectable subtype, with levels significantly higher than G1 (P = 0.001), G3 (P = 0.0127) or G4 (P = 0.002).

Essentially, these results indicated that serum antibodies against MtM or Acr were predominantly IgG (subtype IgG2) and its levels in HCWs and ATB patients did not differ significantly. Nonetheless, compared with ATB, the HCWs exhibited a significantly higher level of anti-MtM IgM and a significantly lower level of anti-Acr IgA antibodies.

## IgG antibody levels against Acr were high or low in relation to the levels against MtM

A weak, though statistically significant correlation was seen between IgG responses to MtM and Acr (n = 53, r = 0.27, P = 0.0494; Fig 3A). Remarkably, in a proportion of subjects Acr produced a higher level of antibody than did MtM (Fig 3B) despite the former being only one of the many antigenic constituents of the latter [8, 12]. The proportion of subjects showing such a high (> 1) $\Delta OD_{Acr}/\Delta OD_{MtM}$ ratio was as follows: OC, 27%; HC, 37.5%; CTB, 20% and ATB, 10% (Fig 3B). These results (particularly the trend between HC and ATB) suggested that the ratio of anti-Acr to anti-MtM IgG could decline with escalating exposure to the infection.

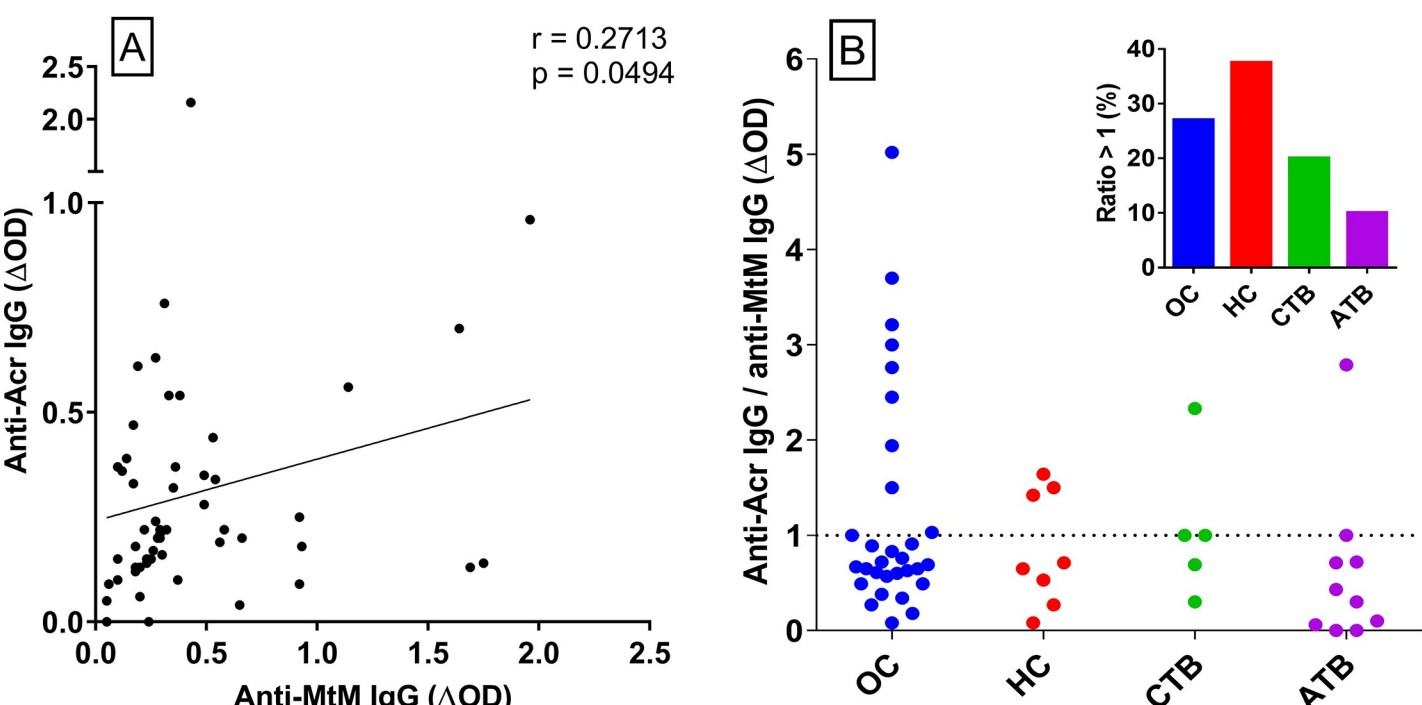

**Fig 3. Relationship between IgG antibodies to Acr and MtM.** Panel A shows Spearman's correlation (n = 53, r = 0.27, P = 0.0494) between two antibody levels ($\Delta OD_{Acr}$ and $\Delta OD_{MtM}$). Panel B shows the ratio $\Delta OD_{Acr}/\Delta OD_{MtM}$ in OC (n = 30), HC (n = 8), CTB (n = 5) and ATB (n = 10) subjects. Dotted line separates subjects showing a ratio of ≤1 and >1. Inset figure depicts proportion of subjects in each category showing a ratio of >1.

## Avidity of anti-MtM IgG in ATB was lower but not significantly different from HCWs

Avidity of IgG antibodies against Mtb 'cell-surface' antigens has also been considered as a bio-marker which could discriminate between active disease and latent TB infection [17, 38]. We therefore determined the avidity of anti-MtM (IgG) in all 10 ATB patients and 10 HCWs (OC) who had comparable antibody levels. Though the median avidity in ATB (AI = 41.75) was lower than that in OC (AI = 54.5,), the difference was not statistically significant (P = 0.1903) due mainly to a wider spread of data in case of ATB (IQR = 20.22–59.51 for ATB vs. 40.95–61.68 for OC) (Fig 4).

## Serum immunoglobulins got bound non-specifically to 'blocked' ELISA plates

To our surprise, the calculated ΔOD value for anti-Acr IgM antibodies in most study subjects was < 0 (Fig 1B and Table 1). This happened because OD values of buffer-coated control wells were higher than the antigen-coated wells. To probe this phenomenon, we compared the

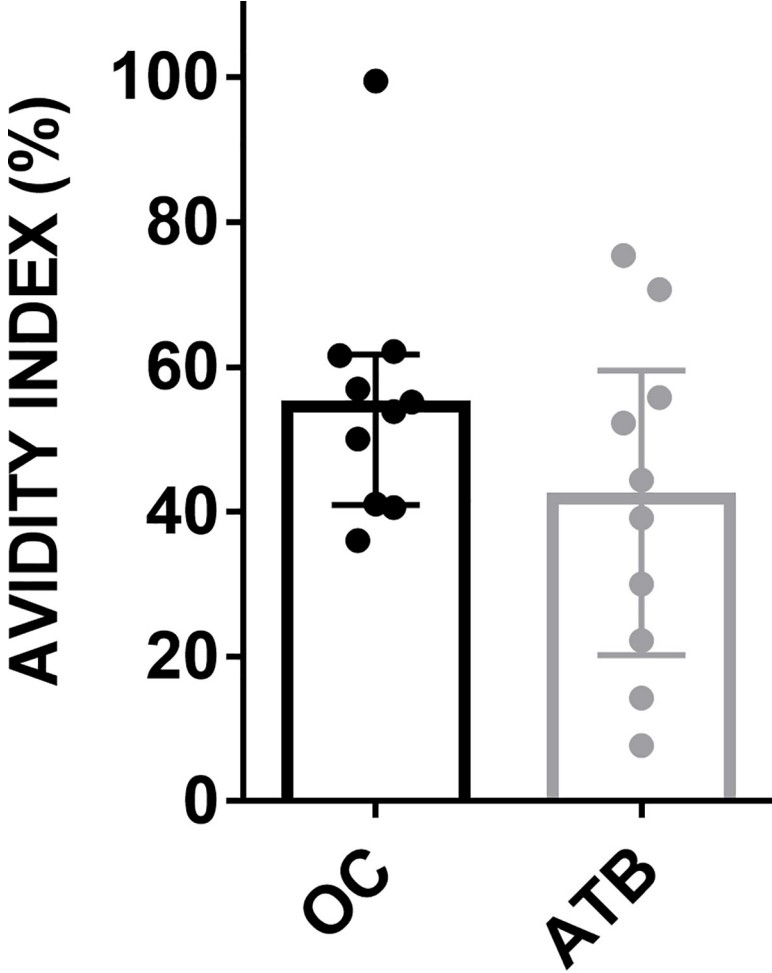

**Fig 4. Avidity index of anti-MtM (IgG) antibodies present in HCW (OC) and ATB sera.** Individual AI values, along with median (bar height) and IQR are shown.

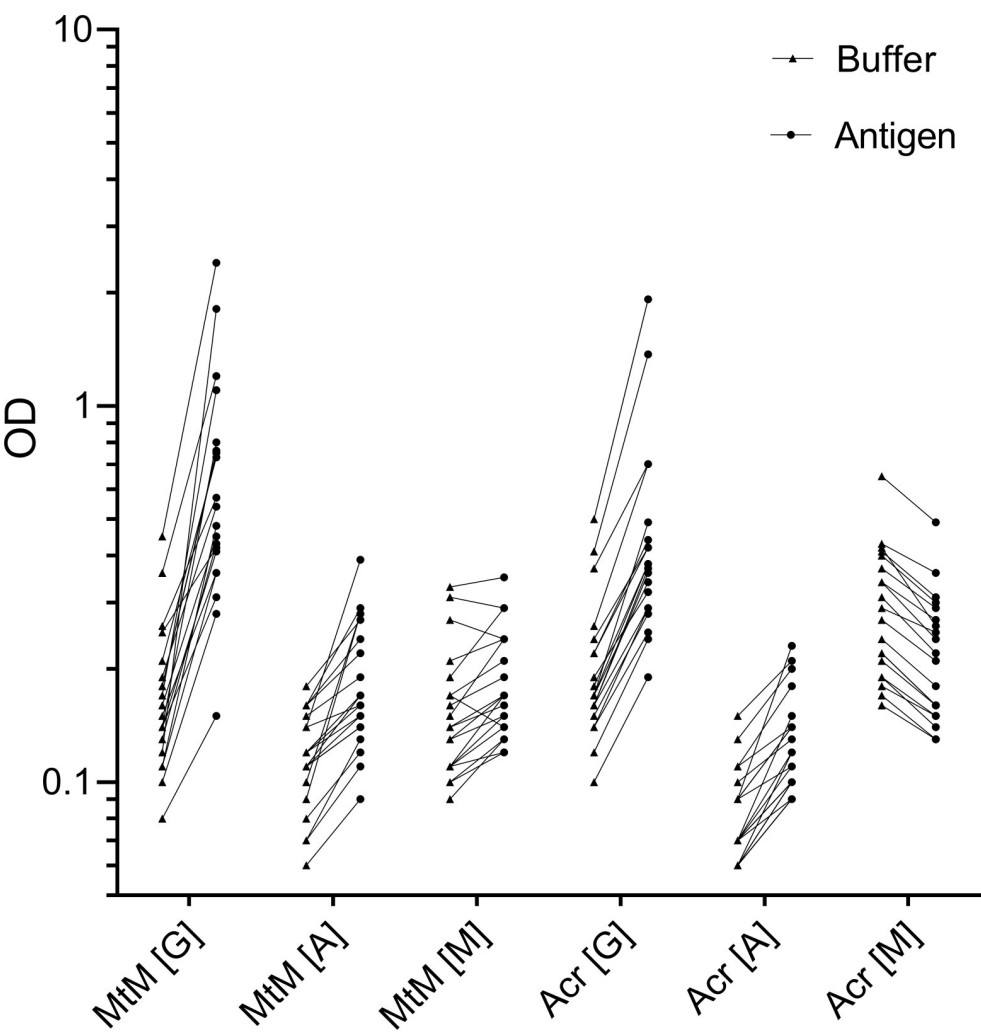

**Fig 5. Non-specific binding of serum immunoglobulins to ELISA plates blocked with milk.** Binding (OD) of IgG, IgA and IgM to ELISA plate-wells coated with buffer (coating buffer) or antigen (MtM or Acr) are shown for 20 sera. Individual pairs of values are connected by solid lines.

extent of non-specific binding for all 3 antibody classes. Fig 5 shows the bindings of serum IgG, IgA and IgM to antigen- as well as buffer-coated ELISA wells. The largest difference in OD pairs was seen for IgG (median values for buffer- and antigen-coated wells, respectively, were 0.16 and 0.51 for MtM; and 0.17 and 0.37 for Acr), followed by IgA (0.12 and 0.17 for MtM, 0.07 and 0.12 for Acr) and IgM (0.14 and 0.17 for MtM, 0.28 and 0.22 for Acr). All differences in OD pairs were statistically significant ($P = 0.0026$ for anti-MtM IgM, and $\leq 0.0001$ for the rest).

While these results show that the test sera (at least at the used dilution) lacked detectable amounts of anti-Acr IgM, they also suggest that coating of ELISA wells with Acr blocked the non-specific binding of IgM more effectively than did milk. This phenomenon (blocking of IgM) may however not be seen with other Mtb antigens, as most study subjects (except ATB) showed a net positive ($> 0$) $\Delta$OD value for anti-MtM IgM (Fig 1A, Table 1).

We also compared the blocking efficiencies of milk and bovine serum albumin in a subset of sera, which were found to be similar.

## Compared to ATB a higher proportion of HCWs showed positive T cell response to Acr

Representative T cell proliferative responses of HCWs and ATB patients towards Acr are depicted in Fig 6. Culture medium alone served as negative control and PHA (a T cell mitogen) was used as a positive control. A relatively weak response to PHA in ATB represents the state of generalized depression in T cell responsiveness in a proportion of such patients [10].

Fig 7 shows individual T cell responses of the study subjects. Proportion of responders (showing a proliferative response above the cut-off for positivity) among HCWs was as follows: OC, 35%; HC, 62.5%; and CTB, 20%. The corresponding value for ATB (10%) was significantly lower than HC (P < 0.05). These results suggest that, akin to its 'proportional' antibody levels (Fig 3B), proliferative T cell responses to Acr could also adopt a declining trend following increase in the burden of infection.

## Enhanced exposure to the infection may also enhance concordance between antibody and T cell responses to Acr

Despite a lack of correlation between T cell and antibody responses of HCWs towards Acr, some concordance was seen between the two responses, particularly in subjects of HC and CTB categories (S5 Fig). Three out of 5 HCs and the lone CTB who showed a positive T cell response to Acr also had a high proportional antibody level ($\Delta OD_{Acr}/\Delta OD_{MtM} > 1$) against Acr. On the other hand, this overlap was present in only 2 out of 10 OCs. This observation, though not statistically significant, suggested that a higher exposure to infection could enhance concordance between antibody and T cell responses to Acr.

## Antibody or T cell responses to Acr and MtM were not affected by responsiveness to tuberculin or vaccination with BCG

There are conflicting reports on association between immune responses to Acr and reactivity to tuberculin or vaccination with BCG [21–24]. However, in the present cohort neither antibody nor T cell responses to Acr or MtM appeared affected by these variables. Both responses did not differ significantly between HCWs who were positive or negative for TST or BCG scar (Fig 8). Earlier, we had shown that their T cell responses to MtM were also not affected by TST or BCG [10].

## Proliferative T cell response to MtM antigens could serve as a sensitive marker for LTBI

A significant correlation (r = 0.60, P < 0.0001; Fig 9) was noted between proliferative T cell responses of HCWs to Acr (this study) and MtM (reported earlier by us, [10]). Similar to the trend witnessed for Acr (Fig 7), T cell responses to MtM had also declined with increasing burden of infection. However, despite these similarities, only 15 (38.5%) HCWs showed a positive response for Acr as compared to 36 (92%) who had shown positivity for MtM (P < 0.0001). Altogether, these results suggest that the T cell response to MtM could serve as a more sensitive marker for LTBI.

## Discussion

Reactivation of LTBI involves resumed multiplication of the dormant bacilli, which could also involve changes in their expressed antigens [7]. Indeed, Mtb overexpresses a characteristic set of genes and proteins, prominent among them being Acr, during the latent phase of infection [18–20, 39]. As a corollary to this the immune responses to Mtb may be considered as

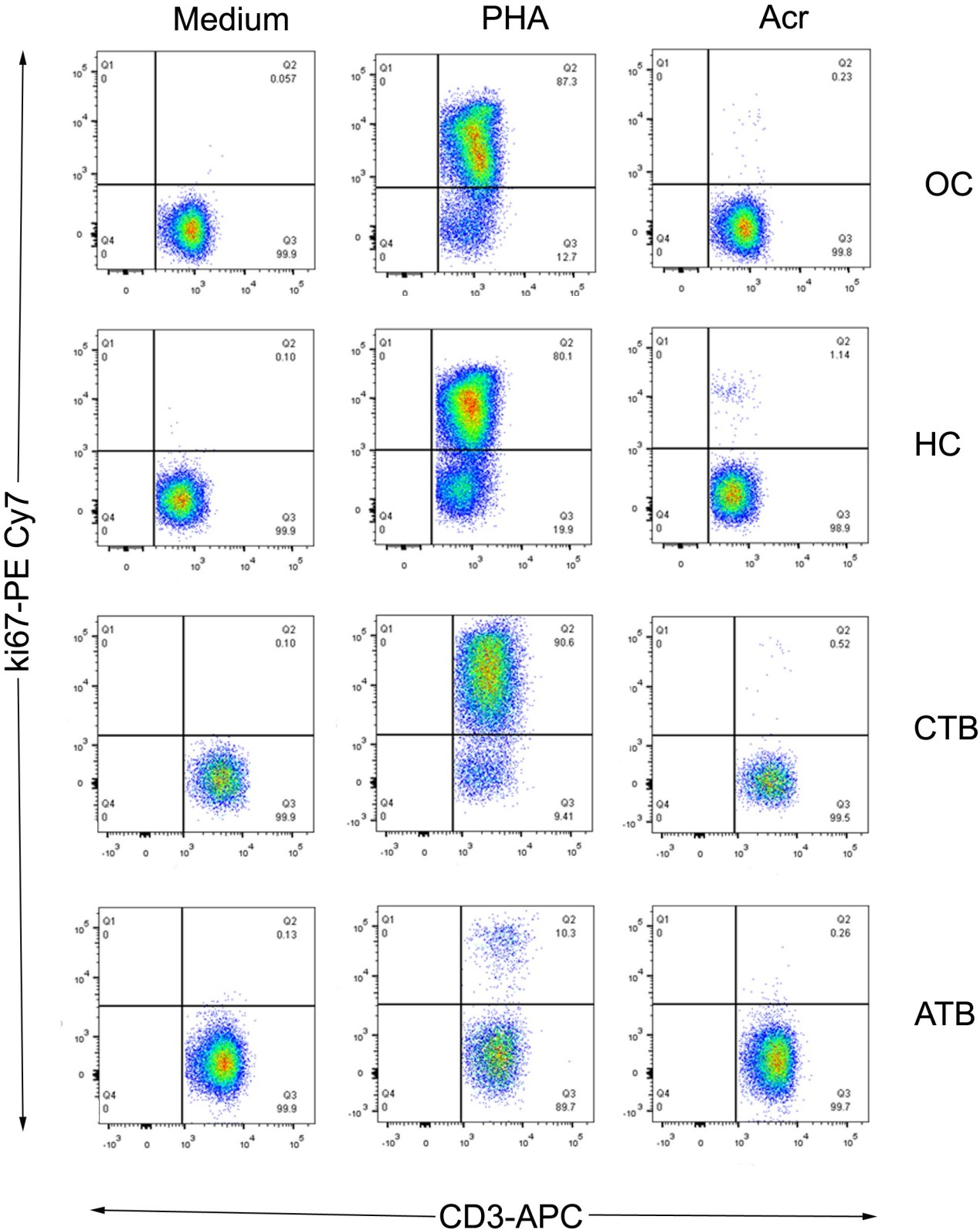

**Fig 6. Representative flow plots of proliferative T cell responses to culture medium, PHA and Acr in HCWs (OC, HC and CTB) and ATB patients.** Proliferating T cells (CD3+Ki67+) are located in the upper right quadrant of each plot.

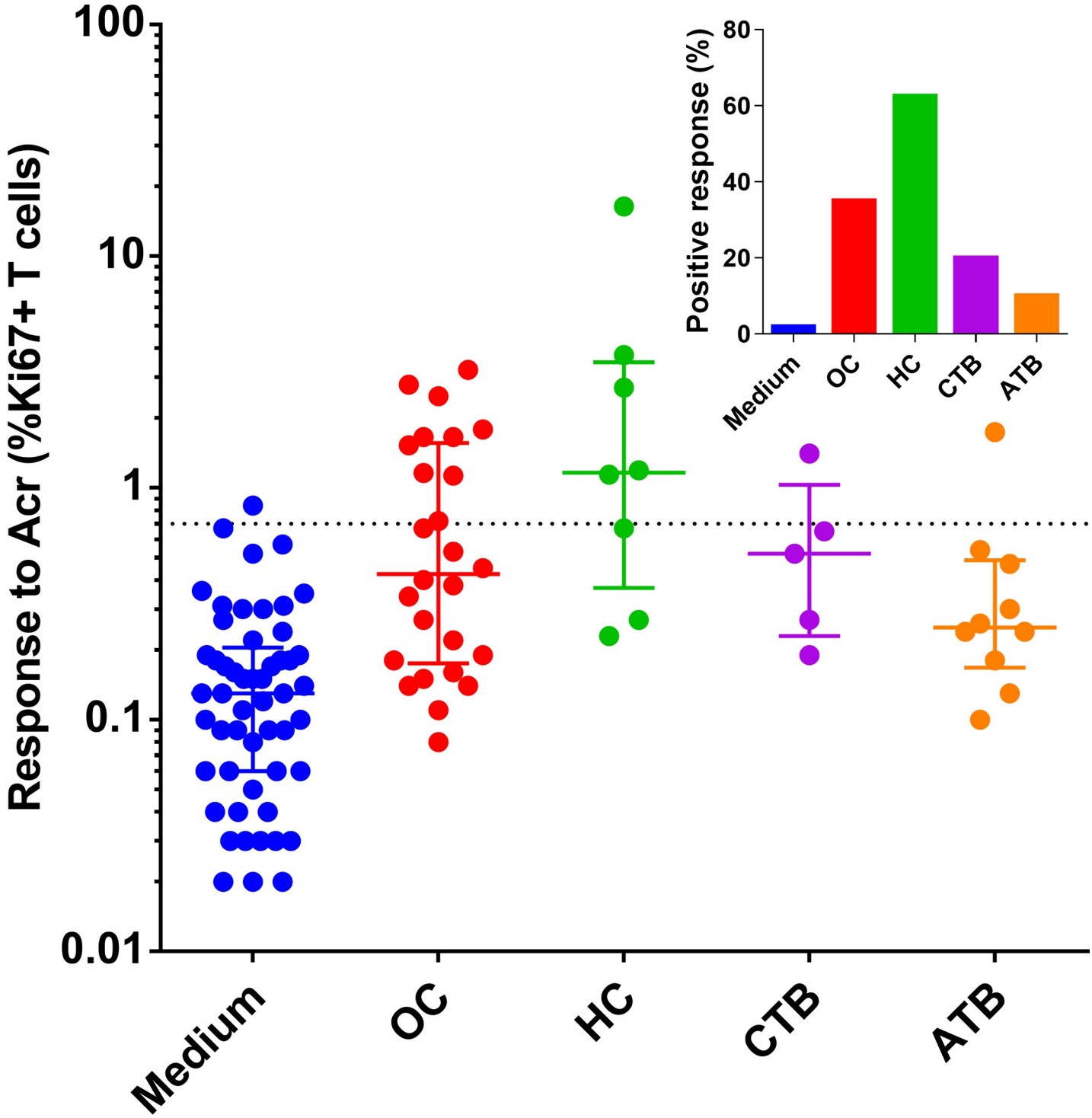

**Fig 7. Proliferative T cell responses of HCWs (OC, HC and CTB) and ATB patients to Acr.** Columns show individual responses (% CD3+Ki67+ cells) along with median and IQR. The proliferative responses to medium alone comprise all study subjects (n = 53). Dotted horizontal line denotes cut-off for a positive response (see Materials and Methods). Inset figure shows % responders in each category (P < 0.05 for the difference between HC and ATB).

dynamic, providing a basis for the discovery of novel biomarker(s) of disease progression [21, 22, 30, 40].

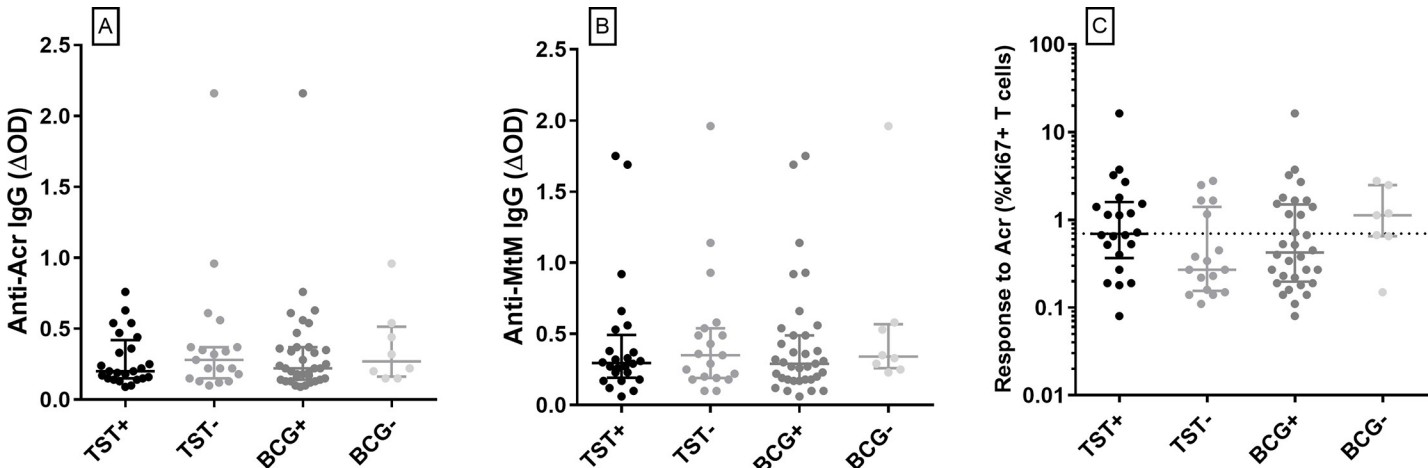

**Fig 8. Antibody and T cell responses to Acr and MtM were not affected by the presence or absence of reactivity towards TST or BCG scar.** Panels A-B show antibody responses (ΔOD) to Acr and MtM, and panel C shows proliferative T cell response (% CD3+Ki67+ cells) to Acr in 43 HCWs (pool of OC, HC and CTB). 24 HCWs were positive (+) and 19 negative (-) for TST. The BCG scar was present (+) in 35 and absent (-) in 8 HCWs. Differences between TST+/- or BCG +/- groups were not significant (all P values > 0.05). Dotted horizontal line in panel C depicts cutoff for a positive T cell response.

Serum antibodies to MtM and its constituent protein Acr were predominantly of IgG class (subclass IgG2), which is consistent with some earlier reports [14, 15, 21]. However, IgG levels against either antigen were unable to differentiate between HCWs and ATB patients. Any prior comparable study on anti-MtM IgG antibodies is not available, whereas available studies on the anti-Acr IgG antibodies have reported equivocal findings. According to some, their levels are low in ATB and high in LTBI [24, 28, 30] and, according to some others, these levels are high in ATB and low in LTBI [21, 25, 29, 41]. While demographic differences in study populations could have led to such diverse results, differences in antibody assay protocols (discussed below) may also have contributed to it.

Compared with ATB, all 3 categories of HCWs (OC, HC and CTB) showed a significantly higher level of anti-MtM IgM and a significantly lower level of anti-Acr IgA antibodies. Though there is no earlier report on the discriminatory potential of anti-MtM IgM antibodies, a study on antibodies to culture-filtrate antigens of Mtb in patients and contacts belonging to the same geographical region (Pakistan) had shown similar results [42]. The authors had reported that, as compared to healthy controls, contacts of TB patients (having cavitory lung disease) had significantly higher levels of IgM antibodies; and concluded that intense exposure of the contacts to Mtb may have led to activation of the innate immune system including IgM antibodies. Polymeric IgM is believed to contribute towards initial defense against invading pathogens until a more specific adaptive immune response takes over. In a model of malaria, IgM+ memory B cells (MBCs) were early responders and dominant producers of antibody-secreting cells [43]. Such MBCs constitutively express toll-like receptors (TLRs) and can proliferate and secrete IgM upon stimulation with TLR-ligands such as cell-wall glycolipids of Mtb. With respect to anti-Acr IgA antibodies, several past studies have highlighted their diagnostic potential. Similar to our findings, Awoniyi et al [44] had also found a significantly higher level of these antibodies in TB patients (compared to LTBI) along with a large area under ROC curves (0.93, in our case it was 0.83). Further, like us, they had also observed a significant decline in their levels after successful treatment. Some other studies highlighting the diagnostic potential of anti-Acr IgA antibodies have been conducted by Abebe at al [29], Davidow et al [30], Legesse et al [41] and Talavera-Paulin et al [45]. Collectively, these studies emphasize the

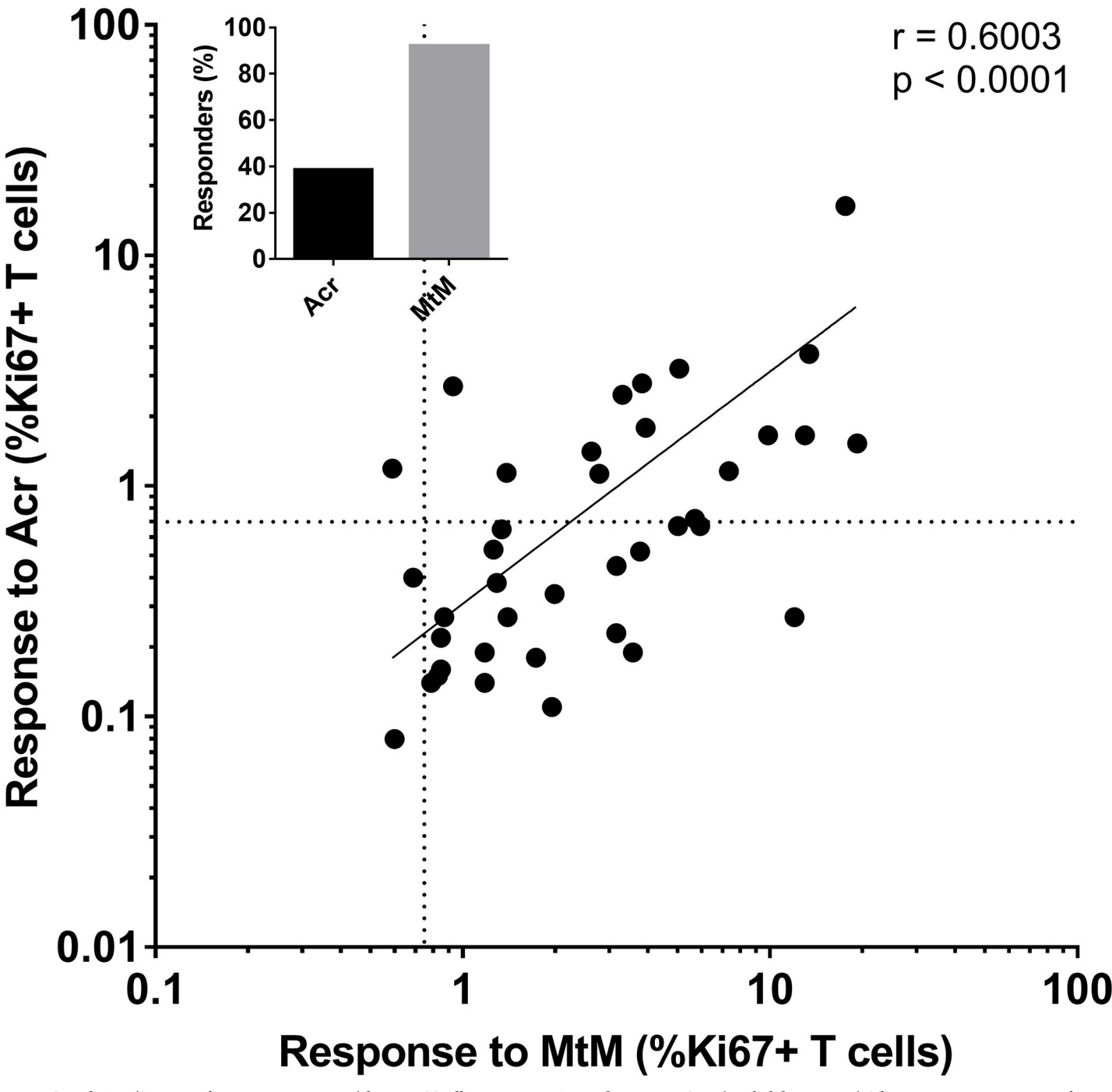

**Fig 9. Correlation (Spearman's r = 0.60, P < 0.0001) between T cell responses to Acr and MtM in HCWs (pooled data, n = 39).** The responses to MtM were taken from our earlier report [10]. Dotted lines represent cutoffs for a positive response to either antigen. Inset figure depicts % responders for each antigen.

need to further explore the potential of anti-MtM IgM and anti-Acr IgA antibodies for the discrimination between HCWs and ATB.

In a proportion of study subjects the levels of IgG antibodies against Acr were higher than those against MtM, despite the fact that Acr is only one of the several antigenic constituents of MtM [8, 12]. More interestingly, the proportion of subjects showing a high (>1) $\Delta OD_{Acr}$/

$\Delta OD_{MtM}$ ratio declined with increasing severity of the infection. This observation is consistent with the reported excessive production of Acr by the dormant bacilli and its reversion to normal levels upon resumption of their exponential growth [18]. It also finds support in some prior studies which have shown that anti-Acr IgG levels are lower in active TB than in LTBI [24, 28, 30]. It could therefore be worthwhile to determine whether a decline in the ratio of anti-Acr to anti-MtM antibodies signals reactivation of LTBI.

We also compared the avidity of anti-MtM IgG antibodies in the sera of HCWs and ATB patients, since it has been proposed that the difference in their avidities could help discriminate between ATB and LTBI. We found that the avidity of antibodies in ATB were lower than that in HCWs. However, this difference was not statistically significant due to a wide variation in avidity in the patients. Earlier studies on the avidity of IgG antibodies against Mtb 'cell-surface' antigens have shown contradictory results. Perley et al [17] had shown a lower avidity, whereas Arias-Bouda et al [38] had shown a higher avidity of these antibodies in ATB patients compared to LTBI. Such variation in results is not entirely unexpected, considering the known heterogeneity in antibody responses of patients [46]. In addition, these variations could also have been caused by differences in the used protocol for avidity determinations [5, 17, 38].

Binding of serum Igs to plastic surfaces is a well known phenomenon, the reason for which is rather obscure [47]. Since none of the blocking agents completely block this non-specific adsorption, it is considered essential to determine the binding of Igs to antigen-coated as well as buffer-coated ELISA plate-wells so as to deduce antigen-specific bindings. Indeed, there are examples where OD values of buffer-coated wells were equal to or even higher than the antigen-coated wells [48, 49]. We encountered a similar phenomenon in this study, which resulted in 'negative' ($< 0$) $\Delta OD$ values for anti-Acr IgM antibodies. This observation was in complete contrast to some of the earlier studies which have shown high levels (OD $>1.5$) of anti-Acr IgM antibodies in ATB patients as well as healthy contacts [29, 32, 45, 50]. Apart from the demographic differences in study populations, such discrepant results may also occur due to purely technical reasons. Determination of non-specific Ig bindings is frequently given a miss by investigators and corresponding controls are not even included in the commercial ELISA kits. False positive results are a major concern for serological antibody assays and the problem is worsened with the use of lower serum dilutions so as to achieve higher assay sensitivities. Lack of SOPs for ELISA has been flagged repeatedly [51] and is the main reason behind their descent as a diagnostic tool for TB [52].

Compared with ATB, a higher proportion of HCWs was positive for the proliferative T cell response to Acr. Indeed, this proportion was significantly high in HC. These results are consistent with those reported for British [21] and African [22] cohorts wherein strongest T cell responses to Acr were seen in latently infected healthy individuals and the responses of TB patients were significantly low. TB patients in the British cohort were further classified according to severity of the infection which revealed its inverse relationship with the response to Acr. Based on this observation, the authors concluded that containment of Mtb infection may partly be mediated by T cells responding to Acr [21]. Likewise, our results also suggest that the T cell response to Acr may adopt a downward trend with increasing exposure to the infection. With respect to association between T cell and antibody responses to Acr, both did coexist in a small proportion of subjects- particularly those who had elevated exposure to infection. This finding was also consistent with that of Wilkinson et al [21] and underscores the emerging view that containment of Mtb infection is a finely orchestrated phenomenon wherein both T cells and antibodies are likely to play complementary roles [14].

The T cell responses of HCWs towards Acr correlated significantly with those towards MtM (reported earlier by us [10]). Even so, a significantly higher proportion of them had shown a positive response to MtM suggesting that it could serve as a more sensitive marker of

LTBI. Several studies in the past have reported that the T cell response in subjects with LTBI is primarily directed at MtM-associated antigens [10–12]. In our previous study [10], it also closely reflected the prevalence of LTBI in India which, according to a study based on T cell responses to Mtb antigens ESAT6, CFP10 and PPD, was found to be over 90% [53]. This estimate may appear alarmingly high but is commensurate the fact that India is home to over a quarter of TB patients worldwide, nearly a million of whom may be 'missing' from the records [1].

We did not see any effect of reactivity towards tuberculin or vaccination with BCG on the antibody or T cell responses to Acr, which is in agreement with some earlier reports [22, 23]. Though in some other studies [21, 24] the BCG vaccinees had shown a positive T cell or antibody response to Acr, the possibility of such persons being exposed to Mtb was not ruled out. Earlier, we had shown that the T cell response to MtM was also not influenced by tuberculin reactivity or BCG vaccination [10].

A relatively small sample size may be considered as a limitation of this study as it could potentially dilute the robustness of drawn conclusions. Even so, these results generate a reasonable degree of confidence since key findings are well-supported by earlier studies. Another important limitation of the study is that, being cross-sectional nature, it did not attempt to determine the predictive value of the described biomarkers. Here it is worth recalling that the primary objective of this study was to comparatively evaluate some of the potential biomarkers with the view to narrow down the choices rather than choosing any one of them. It is well recognized that, due to the low rate of reactivation of LTBI over a protracted period of time, validation of any new biomarker would require a long prospective follow-up of a much larger cohort [2].

To conclude, this study has evaluated the immune responses to Mtb membrane (MtM) and the membrane-associated antigen Acr for detection as well as prediction of the outcome of LTBI in a TB hyperendemic setting. With respect to the value of antibody responses, IgM and IgA class of antibodies to MtM and Acr showed a much greater promise than corresponding IgG antibodies. However, it must be emphasized that the antibody levels reported in this study (and also in most of earlier studies) are 'semi-quantitative' in nature, hence may lack the required degree of robustness. For the present, it appears prudent to focus attention on the T cell responses which also find good support from earlier studies. In this respect, this study draws attention, apart from Acr, to MtM as a rich source of antigens which could serve as more reliable predictors of LTBI. However, a more extensive study of Mtb membrane antigens would give a better understanding to select the correct antigens for these assays.

## Supporting information

**S1 Text. Purchased materials.**
(PDF)

**S1 Fig. Avidity determination for anti-MtM antibodies.**
(PDF)

**S2 Fig. Gating strategy.**
(PDF)

**S3 Fig. Representative data on viability and proliferative capacity of T cells.**
(PDF)

**S4 Fig. Levels of IgG subclasses against MtM.**
(PDF)

**S5 Fig. Concordance between T cell and antibody responses to Acr.**
(PDF)

**S1 File. Raw data for figures and table.**
(PDF)

**S1 Table. Significance of differences (P values) in antibody levels.**
(PDF)

# Acknowledgments

We are grateful to the host Institute (SGPGIMS, Lucknow) for laboratory support. SS is an Emeritus Scientist and SKK is a Senior Research Fellow of Indian Council of Medical Research.

# Author Contributions

**Conceptualization:** Sudhir Sinha.

**Data curation:** Shashi Kant Kumar.

**Formal analysis:** Sudhir Sinha.

**Investigation:** Suvrat Arya, Prerna Kapoor, Alok Nath, Ramnath Misra.

**Methodology:** Shashi Kant Kumar.

**Project administration:** Ramnath Misra.

**Resources:** Amita Aggarwal, Ramnath Misra, Sudhir Sinha.

**Supervision:** Amita Aggarwal, Sudhir Sinha.

**Validation:** Amita Aggarwal, Sudhir Sinha.

**Visualization:** Shashi Kant Kumar, Sudhir Sinha.

**Writing – original draft:** Sudhir Sinha.

**Writing – review & editing:** Amita Aggarwal.

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
