## [Decision Letter · Decision Letter 0]

29 Oct 2019

PONE-D-19-27417

Immune responses to Mycobacterium tuberculosis membrane-associated antigens including alpha crystallin can potentially discriminate between latent infection and active tuberculosis disease

PLOS ONE

Dear Dr Sinha,

Thank you for submitting your manuscript to PLOS ONE. After careful consideration, we feel that it has merit but does not fully meet PLOS ONE’s publication criteria as it currently stands. Therefore, we invite you to submit a revised version of the manuscript that addresses all the points raised during the review process in a point by point manner. This is essential in order to ensure that the manuscript is technically sound, and that the data presented support the conclusions. 

We would appreciate receiving your revised manuscript by Dec 13 2019 11:59PM. To enhance the reproducibility of your results, we recommend that if applicable you deposit your laboratory protocols in protocols.io, where a protocol can be assigned its own identifier (DOI) such that it can be cited independently in the future. For instructions see: http://journals.plos.org/plosone/s/submission-guidelines#loc-laboratory-protocols

We look forward to receiving your revised manuscript.

Kind regards,

Katalin Andrea Wilkinson, PhD

Academic Editor

PLOS ONE

Journal Requirements:

Reviewers' comments:

Reviewer's Responses to Questions

**Comments to the Author**

1. Is the manuscript technically sound, and do the data support the conclusions?

Reviewer #1: Partly

Reviewer #2: Partly

2. Has the statistical analysis been performed appropriately and rigorously? 

Reviewer #1: No

Reviewer #2: Yes

3. Have the authors made all data underlying the findings in their manuscript fully available?

Reviewer #1: Yes

Reviewer #2: Yes

4. Is the manuscript presented in an intelligible fashion and written in standard English?

Reviewer #1: Yes

Reviewer #2: Yes

5. Review Comments to the Author

Reviewer #1: The authors have investigated antibody responses to alpha-crystallin and an Mtb cell membrane fraction in a range of individuals with presumed varying exposure to Mtb, as well as active TB and cured TB.

Major comments

1. Clinical cohort description:

i. How was ‘cured TB’ defined?

ii. Presumably none of the patients with active TB were culture positive, and no drug sensitivity testing was performed?

iii. From the referenced paper describing the cohort, it appears that several of the participants in the active TB group were on antitubercular therapy. Is this the case for the samples interrogated here? Although <3 weeks, this is still sufficient time for sputum conversion to occur.

iv. What is the distribution of TST positivity in the HCW groups? For example, groups are divided into OC and HC, but how many of each of these groups are TST+?

v. Why is the data rather not represented then as TST- HCW, and TST+ HCW, as these groups may be more immunologically distinct?

vi. What is meant by ‘suspected, or had tested for HIV?’ How many were tested, and what were the results? If HIV status is unknown, then this should be simply stated.

vii. If the assumption is that the intensity of exposure is greater in previous household contacts than occupational contacts, is the duration of time since exposure known?

2. Methods:

i. The concentration of secondary antibodies should be given in the methods.

2. Statistical analysis

i. Student’s t-test was used with the presumption of normal data distribution. What is the evidence for selecting a parametric test for this ELISA dataset?

ii. For multiple groups, statistical tests with multiple comparison should be used.

iii. It is not indicated in the figure legends where each statistical method listed is applied. For example, for the correlations, no statistical test is mentioned.

3. Figures:

i. The figures supplied are too low resolution to be clearly read. Please provide higher resolution images of figures.

ii. The figure legends would be improved by stating the statistical methods, numbers and quantitative results obtained. It would also make the figures clearer to interpret if statistical significance was included in the figures.

4. Results:

a) Figure 1:

i. Are there any significant differences using multiple comparisons for each Ig, across the 4 groups and 2 antigens?

ii. How was the inference made that IgG was the predominant isotype? The assays used appear semi-quantitative, a limitation of ELISA without a standard curve. They also utilize different secondary antibodies at different concentrations.

b) Figure 2

i. The conclusion that non-specific serum binding is ‘strikingly’ greater than antigen binding for IgM appears to be a technical challenge rather than biological.

ii. The results that IgM are higher in background wells than antigen-coated wells suggests that IgM is not detected at that concentration of sera, and should be stated as such.

iii. Comparing blocking buffers is reasonable, however this did not resolve the issue. Further optimization of antigen concentration, serum and secondary dilution could improve detection specific IgM.

ii. Prior to describing this as a “striking” finding with “unknown mechanism”, could the authors please review the optimization of this IgM assay, or perhaps include positive controls of known IgM responder serum.

c) Figure 3A and 3B:

i. Please revise figures to include statistical tests, p value and numbers.

ii. How is the lack of correlation to be understood, given that Acr is likely a component of the more immunogenic MtM antigen fraction?

iii. Can the ratio of OD be meaningfully interpreted, given that the OD is a semi-quantitative measure, and MtM is at a higher concentration? It could be understood that these ratios are from technical aspects of the assay rather than biological.

b) Figure 4:

i. The avidity experiment is an interesting approach, given observations in a recent publication (Lu et al, IFN-γ-independent immune markers of Mycobacterium tuberculosis exposure, Nature Medicine, 2019).

ii. What is the specific p value on this figure?

iii. Has a concentration curve of urea, per the above article, been tested? Perhaps this could reveal avidity differences in the contacts vs active TB group.

a. It would be interesting to see if an inverse trend existed for cytosolic antigen fraction.

d) Figure 5 and 6:

i. A representative complete gating strategy should be shown, including a live-dead stain.

ii. The positive population for Ki67 seems to be placed quite high and underestimate stimulation, especially in the active TB panel.

iii. The gate should be placed closer to the negative population with gating derived from a secondary antibody FMO panel shown in the figure. Could the authors please show an FMO figure where this gating was validated? This may even improve the results of the assay.

iv. Could the authors show the raw data analysed as frequency of live Ki67+ T cells.

v. It is unclear what is on the Y-axis. What is meant by ‘responder T cells?’ The figure insert is also illegible.

vi. For the stimulation in this assay, is it possible that more antigen-specific cells are present in ‘cured TB’ and active TB, but these cells die during the 5 day stimulation?

e) Figure 7:

i. It is unclear which samples are these – are they all the OC and HC samples pooled?

ii. The axes do not include antigens tested.

iii. What are the numbers in each group?

iv. What is the cut-off for TST, and how was this determined if HIV infection was not determined?

f) Figure 8:

i. What is the concentration of MtM used in the stimulation assay? Is it fair to conclude that more cells respond to MtM when perhaps non-saturating levels of Acr were used, and the concentrations differed?

Minor comments

1. Single quotation marks appear numerous times in the article, perhaps to make the point that some of these notions are contentious in the literature. These could serve to be removed and the point made simply for readability. Ie. it is not needed to put ‘blocking’ buffer in quotation marks for example.

2. What is meant by a front-line antigens?

3. The authors repeatedly preface their results with several citations and literature review in the results section. This information may be more appropriate in the discussion section.

4. The language includes colloquialisms such as ‘turned out to be’ that seem inappropriate per standard academic convention.

5. Could the authors please revise for grammatical errors.

Reviewer #2: In recent years there are several reports about immunological methods to identify TB infected people and to differentiate active TB from latent TB infection. This paper analyze the antibody response as well as T cell response against M. tuberculosis membrane-associated antigens and the protein Acr.

Introduction is well written and enough information has been included to understand the problem. I consider the methods were designed and performed adequately. Still, I have some general comments that authors should consider to improve the article.

Although he actual immunological methods cannot discriminate between active TB and latent TB, the recognized assay to be include, in all works testing new methods, is the Quantiferon Gold as a reference assay. Authors must include the reason why they did not use it and discuss how this could affect the results they found. I mean, how would they compare how good is the method they propose against a recognized one. Also it would be good if they have had the quantiferon data for each patient for a better group distribution. They neither did it in the previous work in reference 9.

Two preparations were tested in this work: membrane-associated antigens and Acr protein. Authors suggest than membrane antigens could be a better marker than Acr for latent TB. However, the 82 antigens identified in the previous study as immunogenic were not analyzed for their specificity for Mtb. The question is if there are some cross reactivity with other mycobacteria species and that could confuse the results and make it harder to interpret them. Authors reported that antibodies and T cell response not being influenced by tuberculin or BCG vaccination, even so, it cannot be excluded a cross reactivity with membrane antigens. Also, it turns out that Acr is one of the main protein of the antigens associated to the membrane; then, how do you know that the response is due to other antigens or mainly due to Acr? A more extensive study about Mtb membrane antigens would give a better understanding to select the correct antigens for the assays.

Another emerging question is the number of samples, I think they are not enough to have right conclusions, but this is already mentioned by the authors in discussion.

6. PLOS authors have the option to publish the peer review history of their article (what does this mean?). If published, this will include your full peer review and any attached files.

Reviewer #1: No

Reviewer #2: No

---

## [Author Response · Author response to Decision Letter 0]

10 Dec 2019

RESPONSE TO REVIEWERS’ COMMENTS

Reviewer #1: 

The authors have investigated antibody responses to alpha-crystallin and an Mtb cell membrane fraction in a range of individuals with presumed varying exposure to Mtb, as well as active TB and cured TB.

Major comments

(We have carried out a major revision of the manuscript in view of these, as well as comments from 2nd Reviewer and Editor.)

1. Clinical cohort description:

Query/comment (i). How was ‘cured TB’ defined? 

Response: TB cure was defined as per WHO criteria (WHO/HTM/TB/2013.2).

Q (ii). Presumably none of the patients with active TB were culture positive, and no drug sensitivity testing was performed? 

R: All patients were sputum-smear positive (1+ to 3+) which was used as a criterion for determining active disease (WHO/HTM/TB/2013.2). Drug sensitivity testing was not performed in 8 patients whose clinical history did not suggest drug-resistance. In the remaining 2, where drug resistance was suspected, Xpert MTB/RIF assay was performed (WHO/HTM/TB/2013.16). However, both were found sensitive for RIF.

Q (iii). From the referenced paper describing the cohort, it appears that several of the participants in the active TB group were on antitubercular therapy. Is this the case for the samples interrogated here? Although <3 weeks, this is still sufficient time for sputum conversion to occur. 

R: As mentioned above, all patients were smear-positive (1+ to 3+) at the time of investigation.

Q (iv). What is the distribution of TST positivity in the HCW groups? For example, groups are divided into OC and HC, but how many of each of these groups are TST+? 

R: As shown in our previous report on this cohort (Ref 10), distribution of TST positivity in individual HCW groups was: OC, 13/30; HC, 7/8 and CTB, 4/5.

Q (v). Why is the data rather not represented then as TST- HCW, and TST+ HCW, as these groups may be more immunologically distinct? 

R: The TST+/- (as well as BCG +/-) HCW data presented in the manuscript (Fig 8) as well as in our previous report (ref 10) suggested that these groups may not be immunologically distinct.

Q (vi). What is meant by ‘suspected, or had tested for HIV?’ How many were tested, and what were the results? If HIV status is unknown, then this should be simply stated. 

R: We follow the policy of HIV testing for all newly diagnosed TB patients. Nine out of 10 patients were tested and found seronegative for HIV. In the remaining 1 patient (who was transferred to us from another clinic) also, we did not suspect HIV based on her clinical records.

Q (vii). If the assumption is that the intensity of exposure is greater in previous household contacts than occupational contacts, is the duration of time since exposure known? 

R: As stated in Materials and Methods, the household contacts (HC) had lived with a smear-positive TB patient for at least 3 months (WHO/HTM/TB/2012.9). Duration of tome since exposure was > 1year. 

2. Methods:

Q (i). The concentration of secondary antibodies should be given in the methods.

R: This has now been given.

2. Statistical analysis

Q (i). Student’s t-test was used with the presumption of normal data distribution. What is the evidence for selecting a parametric test for this ELISA dataset? 

R: Thanks for pointing this out. Indeed, the dataset was not normally distributed. We have therefore reanalyzed the data by non-parametric methods (see revised Materials and Methods) and incorporated the required changes in Results and Discussion.

Q (ii). For multiple groups, statistical tests with multiple comparison should be used. 

R: We have now done it and results are shown in Results.

Q (iii). It is not indicated in the figure legends where each statistical method listed is applied. For example, for the correlations, no statistical test is mentioned. 

R: The figure legends have now been modified according to this suggestion.

3. Figures:

Q (i). The figures supplied are too low resolution to be clearly read. Please provide higher resolution images of figures. 

R: High-resolution images were provided to the journal. They can perhaps be accessed though a hot link on top right corner of each image.

Q (ii). The figure legends would be improved by stating the statistical methods, numbers and quantitative results obtained. It would also make the figures clearer to interpret if statistical significance was included in the figures. 

R: The figure legends have been modified as suggested.

4. Results:

a) Figure 1:

Q (i). Are there any significant differences using multiple comparisons for each Ig, across the 4 groups and 2 antigens? 

R: These results are now shown in revised Figure 1.

Q (ii). How was the inference made that IgG was the predominant isotype? The assays used appear semi-quantitative, a limitation of ELISA without a standard curve. They also utilize different secondary antibodies at different concentrations. 

R: The assay is indeed semi-quantitative and we have now mentioned this limitation in revised Discussion. However, the concentration of secondary antibodies was not limiting as determined during the assay optimization. 

b) Figure 2:

Q (i). The conclusion that non-specific serum binding is ‘strikingly’ greater than antigen binding for IgM appears to be a technical challenge rather than biological. 

R: This is possible and we have modified the Results and Discussion accordingly.

Q (ii). The results that IgM are higher in background wells than antigen-coated wells suggests that IgM is not detected at that concentration of sera, and should be stated as such. 

R: Necessary modification has been made in Results.

Q (iii). Comparing blocking buffers is reasonable, however this did not resolve the issue. Further optimization of antigen concentration, serum and secondary dilution could improve detection specific IgM. 

R: We have revised the relevant text in the light of these comments.

Q (iv). Prior to describing this as a “striking” finding with “unknown mechanism”, could the authors please review the optimization of this IgM assay, or perhaps include positive controls of known IgM responder serum. 

R: We have revised the relevant text in the light of these comments. It is however possible that blocking of IgM is a unique property of Acr, since MtM did not do so.

c) Figure 3A and 3B:

Q i. Please revise figures to include statistical tests, p value and numbers. 

R: This revision has been done.

Q ii. How is the lack of correlation to be understood, given that Acr is likely a component of the more immunogenic MtM antigen fraction?

R: We stand corrected. New analysis by non-parametric methods has revealed a significant (though weak) correlation between the two antibody responses. We have revised the Discussion accordingly.

Q iii. Can the ratio of OD be meaningfully interpreted, given that the OD is a semi-quantitative measure, and MtM is at a higher concentration? It could be understood that these ratios are from technical aspects of the assay rather than biological.

R: We agree with this observation and have incorporated it in Discussion.

b) Figure 4:

Qi. The avidity experiment is an interesting approach, given observations in a recent publication (Lu et al, IFN-γ-independent immune markers of Mycobacterium tuberculosis exposure, Nature Medicine, 2019).

R: Thanks for the suggestion. We have also incorporated this paper in revised manuscript (ref 5}.

Qii. What is the specific p value on this figure? 

R: This has now been given.

iii. Has a concentration curve of urea, per the above article, been tested? Perhaps this could reveal avidity differences in the contacts vs active TB group.

R: In absence of a ‘gold standard’ method for avidity determination, the protocol used by us (ref 36) has been used more commonly (recently in Nature Communications, 2019, 10:2174). Though some workers have used thiocyanate, it (being a biohazard) is being replaced by urea. Nonetheless, following your suggestion we also tried the protocol described by Lu et al. In doing so, besides using the pooled sera (as done by them) we also used individual HCW and ATB sera. However, 50% inhibition was not attained in 5 out of 6 sera as well as the pooled sera (S2 Fig). This could have happened due to the difference in nature of antigens (PPD vs MtM) or avidity of antibodies.

a. It would be interesting to see if an inverse trend existed for cytosolic antigen fraction.

R: The immunoproteome of Mtb is rich in membrane and not cytosolic proteins (ref 7). We (refs 13, 14) and others (refs 15-17) have shown that human antibodies are directed mainly towards MtM antigens. We also observed that antibody levels against cytosolic antigens are generally low (ΔOD < 0.2) hence did not proceed with this experiment. 

d) Figure 5 and 6:

Q i. A representative complete gating strategy should be shown, including a live-dead stain.

R: A representative gating strategy is shown as S3 Fig. We also determined the viability of T cells (after 5 days of whole blood cultures) using propidium iodide staining and flow cytometry (ref 37). As shown in S4 Fig (data of a smear-positive TB patient), > 90% CD3+ cells were viable on day 6 and 95% of them showed a strong proliferative response to PHA. 

Q ii: The positive population for Ki67 seems to be placed quite high and underestimate stimulation, especially in the active TB panel. 

R: In deciding the position of quadrants, we followed the adage ‘unstimulated control best teaches where to place the quadrants’ (Cytometry Part A, 2006, 69A:1037–1042). Further, since Ki67 positive and negative populations were distinctly apart, moving the horizontal line a bit lower did not achieve any worthwhile gain in positivity of stimulated cells. However, it frequently (and falsely) enhanced the positivity of unstimulated cells. As shown in our previous report (ref 10), the response of patients (as well as HCWs) to PHA was variable. For example, another patient (with 3+ smear positivity) showed a strong response (S4 Fig).

Q iii: The gate should be placed closer to the negative population with gating derived from a secondary antibody FMO panel shown in the figure. Could the authors please show an FMO figure where this gating was validated? This may even improve the results of the assay.

R: We chose Biological Comparison Control over FMO control for the following reasons.

1. FMO controls are more relevant in multicolor (> 4 colors) experiments (Cytometry Part A, 2006, 69A:1037–1042). Besides, there are also some limitations of FMO controls. Because it does not contain an antibody in the channel of interest, an FMO control does not provide the measure of background staining when that antibody is actually included. Further, FMO addresses spillover-induced background but not non-specific antibody binding.

2. A biological comparison control is recognized as the most relevant control for determining positivity of the test samples (Cytometry Part A, 2006, 69A:1037–1042). In most cases this control is far more relevant than an isotype control or FMO. Like an FMO control, the unstimulated control accounts for spillover effects on the channel of interest as it includes all antibody conjugates present in the test sample. In addition, like an isotype control, it also accounts for nonspecific staining in the channel of interest.

Q iv: Could the authors show the raw data analysed as frequency of live Ki67+ T cells.

R: T cells were fixed and permeabilized prior to staining for Ki67 hence were not alive.

Qv: It is unclear what is on the Y-axis. What is meant by ‘responder T cells?’ The figure insert is also illegible.

R: This figure has been modified and the inset has been enlarged.

Q vi: For the stimulation in this assay, is it possible that more antigen-specific cells are present in ‘cured TB’ and active TB, but these cells die during the 5 day stimulation?

R: As mentioned above (S4 Fig), over 90% T cells of ATB patients were alive on day-6 and also responded strongly to PHA.

e) Figure 7:

Q (i-iii). It is unclear which samples are these – are they all the OC and HC samples pooled? The axes do not include antigens tested. What are the numbers in each group?

R: This information has now been given.

Q (iv). What is the cut-off for TST, and how was this determined if HIV infection was not determined?

R: The cut-off for TST was 10 mm induration (as per IUATLD and WHO guidelines, discussed in Ref 10). TST was done only in HCWs, none of whom was positive for HIV.

f) Figure 8:

i. What is the concentration of MtM used in the stimulation assay? Is it fair to conclude that more cells respond to MtM when perhaps non-saturating levels of Acr were used, and the concentrations differed?

R: Equal concentrations (5 µg/ml) of both Acr and MtM (ref 10) were used. Moreover, as Acr happens to be just one of the 105 proteins identified by us in the MtM proteome (ref 8), it may not be fair to conclude that non-saturating levels of Acr were used. 

Minor comments

1. Single quotation marks appear numerous times in the article, perhaps to make the point that some of these notions are contentious in the literature. These could serve to be removed and the point made simply for readability. Ie. it is not needed to put ‘blocking’ buffer in quotation marks for example.

R: We concur with this observation and have removed single quotation marks wherever unnecessary.

2. What is meant by a front-line antigens?

R: We have replaced this phrase with ‘potential biomarker’.

3. The authors repeatedly preface their results with several citations and literature review in the results section. This information may be more appropriate in the discussion section.

R: We have now removed most citations from Results. They are however still needed at some places to emphasize/justify the need for undertaking a particular investigation.

4. The language includes colloquialisms such as ‘turned out to be’ that seem inappropriate per standard academic convention.

R: We fully agree with this and have removed this phrase.

5. Could the authors please revise for grammatical errors.

R: We have re-read the manuscript and have made required corrections.

Reviewer #2: 

In recent years there are several reports about immunological methods to identify TB infected people and to differentiate active TB from latent TB infection. This paper analyze the antibody response as well as T cell response against M. tuberculosis membrane-associated antigens and the protein Acr. Introduction is well written and enough information has been included to understand the problem. I consider the methods were designed and performed adequately. Still, I have some general comments that authors should consider to improve the article.

Comment: Although the actual immunological methods cannot discriminate between active TB and latent TB, the recognized assay to be included, in all works testing new methods, is the Quantiferon Gold as a reference assay. Authors must include the reason why they did not use it and discuss how this could affect the results they found. I mean, how would they compare how good is the method they propose against a recognized one. Also it would be good if they have had the quantiferon data for each patient for a better group distribution. They neither did it in the previous work in reference 9.

Response: We have used TST as the reference assay, reasons for which have now been elaborated more explicitly in revised Introduction (2nd paragraph). 

Comment: Two preparations were tested in this work: membrane-associated antigens and Acr protein. Authors suggest than membrane antigens could be a better marker than Acr for latent TB. However, the 82 antigens identified in the previous study as immunogenic were not analyzed for their specificity for Mtb. The question is if there are some cross reactivity with other mycobacteria species and that could confuse the results and make it harder to interpret them. 

Response: We fully agree with this observation. In fact we have not proposed that the crude membrane should be used as such. In concluding paragraph of Discussion we had said that “In addition to Acr, the study draws attention to MtM as an emerging source of immunodominant T cell antigens”. We have now revised the Discussion to further clarify this point. However, in real life the ‘non-tuberculous mycobacteria’ as well as BCG do not seem to confound the results of TST which also uses a crude antigen (Farhat et al, Int J Tuberc Lung Dis, 2006; 10:1192).

Comment: Authors reported that antibodies and T cell response not being influenced by tuberculin or BCG vaccination, even so, it cannot be excluded a cross reactivity with membrane antigens. 

Response: We have revised the relevant text accordingly.

Comment: Also, it turns out that Acr is one of the main protein of the antigens associated to the membrane; then, how do you know that the response is due to other antigens or mainly due to Acr? A more extensive study about Mtb membrane antigens would give a better understanding to select the correct antigens for the assays.

Response: We completely agree with this suggestion and have revised the Discussion accordingly. In fact, we have made some such attempts in the past (ref 8) but have not been able meet any remarkable success. We nevertheless are seized with this problem.

Comment: Another emerging question is the number of samples, I think they are not enough to have right conclusions, but this is already mentioned by the authors in discussion. 

(No response needed).

RESPONSE TO EDITOR’S COMMENTS 

Comment: We note that you have included the phrase “data not shown” in your manuscript. Unfortunately, this does not meet our data sharing requirements. PLOS does not permit references to inaccessible data. We require that authors provide all relevant data within the paper, Supporting Information files, or in an acceptable, public repository. Please add a citation to support this phrase or upload the data that corresponds with these findings to a stable repository (such as Figshare or Dryad) and provide and URLs, DOIs, or accession numbers that may be used to access these data. Or, if the data are not a core part of the research being presented in your study, we ask that you remove the phrase that refers to these data.

Response: We have now removed this phrase from Results section (in context of Fig 3) without giving any additional data as the corresponding observation is trivial in nature. We have also removed this phrase from Discussion, but have given the data (as S7 Fig) since we needed to make an important discussion point.

---

## [Decision Letter · Decision Letter 1]

3 Jan 2020

PONE-D-19-27417R1

Immune responses to Mycobacterium tuberculosis membrane-associated antigens including alpha crystallin can potentially discriminate between latent infection and active tuberculosis disease

PLOS ONE

Dear Dr Sinha,

Thank you for submitting your manuscript to PLOS ONE. After careful consideration, we feel that it has merit but does not fully meet PLOS ONE’s publication criteria as it currently stands. Therefore, we invite you to submit a revised version of the manuscript that addresses the points raised during the review process.

We would appreciate receiving your revised manuscript by Feb 17 2020 11:59PM. To enhance the reproducibility of your results, we recommend that if applicable you deposit your laboratory protocols in protocols.io, where a protocol can be assigned its own identifier (DOI) such that it can be cited independently in the future. For instructions see: http://journals.plos.org/plosone/s/submission-guidelines#loc-laboratory-protocols

We look forward to receiving your revised manuscript.

Kind regards,

Katalin Andrea Wilkinson, PhD

Academic Editor

PLOS ONE

Reviewers' comments:

Reviewer's Responses to Questions

**Comments to the Author**

1. If the authors have adequately addressed your comments raised in a previous round of review and you feel that this manuscript is now acceptable for publication, you may indicate that here to bypass the “Comments to the Author” section, enter your conflict of interest statement in the “Confidential to Editor” section, and submit your "Accept" recommendation.

Reviewer #1: (No Response)

Reviewer #2: All comments have been addressed

2. Is the manuscript technically sound, and do the data support the conclusions?

Reviewer #1: Yes

Reviewer #2: Yes

3. Has the statistical analysis been performed appropriately and rigorously? 

Reviewer #1: (No Response)

Reviewer #2: (No Response)

4. Have the authors made all data underlying the findings in their manuscript fully available?

Reviewer #1: Yes

Reviewer #2: (No Response)

5. Is the manuscript presented in an intelligible fashion and written in standard English?

Reviewer #1: Yes

Reviewer #2: (No Response)

6. Review Comments to the Author

Reviewer #1: The authors have undertaken substantial revision of the manuscript exploring humoral immunogenicity of membrane-associated proteins in human TB. The principle findings include (a) higher levels of IgM in contacts than in active TB patients (b) higher Acr IgA in active TB than contacts (c) altered ratios of IgG to Acr vs Mtm across the spectrum of TB disease.

In general, the comments has been addressed.

Further comments:

1. The major concern of statistical analysis has been resolved with the correction of non-parametric and multiple comparison tests.

2. The rebuttal to the flow cytometry gating is accepted. The live/dead stain at day 6 confirms the cells were still alive in active TB.

3. Thank you for the additional experiments regarding the antibody avidity. It is suggested to put the p-value in the figure given that the trend is discussed. Could the heterogeneity in antibody avidity in active TB be due to different durations of disease prior to presentation? Perhaps something to explore in future studies with greater powering.

4. The finding of IgM being greater in contacts than in active TB is interesting, suggesting the chronicity of active TB drives class-switching. Could a mechanism be included in the discussion?

5. The r-squared and p values should be inserted in the scatter plot figures (correlations).

6. The results regarding IgM not detectable against Acr is likely technical (concentration of Acr coating?), but the authors have addressed this clearly in the discussion. In the way it is presented, this could still be of benefit for the research community.

7. Line 268: Can immunoglobulin levels truly be directly compared on an OD basis given it is being compared across different antigens and secondary antibodies? Would this not require a standard curve of subclasses to derive a specific concentration of immunoglobulin? However, it could be correct to state that IgG and then later for figure 6, IgG2 (line 316), is the most readily detectable immunoglobulins in their given assays.

Minor comments:

1. Line 66: What is meant by ‘presumptive’ in single quotation marks?

2. Line 88: Missing a word – Mtb membrane lysate or proteins?

3. Line 97: Why is ‘Mtb complex’ in single quotation marks?

4. Line 95-96: What is meant by sustain the bacilli in

5. Line 188: Should be IgA not AgA

6. Line 200: The concentration of the secondary should be included.

7. Line 256: Missing word – elaborated ‘on’.

8. Line 394: It is unclear what is meant by quantum of infection increasing.

9. Line 556: Typo in semi-quantitative. The point with semi-quantitative is that levels of antibodies cannot be directly compared when using different reagents (secondaries) or assays. Further, it also poses a challenge for defining cut-off values for diagnostic tests as the units are arbitrary and contingent on the individual assay. However, it is accepted that this study is exploratory in nature.

10. Line 451: Why is dynamic in single quotation marks?

11. Line 492: Why is ‘cell-surface’ in single quotation marks?

12. Line 512: “gets worsened” should perhaps be “is worsened”

Reviewer #2: My comments about some of the responses given by authors

Author’s response: We have used TST as the reference assay, reasons for which have now been elaborated more explicitly in revised Introduction (2nd paragraph).

Authors have explained the reason why they used TST as the reference assay. I may consider this a valid reason, but it would be easier to compare their results with others reports using IGRAs. Still the data provided in this paper may be useful to complement the information about this topic.

Author’s response: We fully agree with this observation. In fact we have not proposed that the crude membrane should be used as such. In concluding paragraph of Discussion we had said that “In addition to Acr, the study draws attention to MtM as an emerging source of immunodominant T cell antigens”. We have now revised the Discussion to further clarify this point. However, in real life the ‘non-tuberculous mycobacteria’ as well as BCG do not seem to confound the results of TST which also uses a crude antigen (Farhat et al, Int J Tuberc Lung Dis, 2006; 10:1192).

I agree with the paragraph mentioned in discussion. But the fact that MtM is a rich source of antigens, most of them unstudied, reveals that it is not close to determine which protein may actually function as biomarker. As author also responded: we have made some such attempts in the past (ref 8) but have not been able meet any remarkable success.

Additional comment:

I expected to see the changes made to the paper highlighted to facilitate the second review, instead I had to go through all the paper to see the changes

7. PLOS authors have the option to publish the peer review history of their article (what does this mean?). If published, this will include your full peer review and any attached files.

Reviewer #1: No

Reviewer #2: No

---

## [Author Response · Author response to Decision Letter 1]

9 Jan 2020

Review Comments to the Author

Reviewer #1: The authors have undertaken substantial revision of the manuscript exploring humoral immunogenicity of membrane-associated proteins in human TB. The principle findings include (a) higher levels of IgM in contacts than in active TB patients (b) higher Acr IgA in active TB than contacts (c) altered ratios of IgG to Acr vs Mtm across the spectrum of TB disease.

In general, the comments has been addressed.

Further comments:

Comment-1. The major concern of statistical analysis has been resolved with the correction of non-parametric and multiple comparison tests.

(No response required)

C2. The rebuttal to the flow cytometry gating is accepted. The live/dead stain at day 6 confirms the cells were still alive in active TB.

(No response required)

C3. Thank you for the additional experiments regarding the antibody avidity. It is suggested to put the p-value in the figure given that the trend is discussed. Could the heterogeneity in antibody avidity in active TB be due to different durations of disease prior to presentation? Perhaps something to explore in future studies with greater powering.

Response: In the cited work [ref 5], authors had determined the avidity of anti-PPD antibodies in terms of molar concentration of urea required to reduce antibody bindings by 50% (IC50). Since they attained IC50 values in their experiment, statistical significance of differences could be calculated by t-test. However, in our case, the IC50 values were not attained (the possible reasons for which had been stated in revised manuscript R1). We nonetheless considered analysing the trends, but the data was insufficient. In any case, the two trends (HCW and ATB) appear comparable (S2 Fig). We had already stated in our previous revision (R1 Discussion) that the known heterogeneity in antibody responses of patients (which could be due to various reasons, including disease duration) and (perhaps more importantly) the differences in used protocol for avidity determinations could have caused these variations in avidity. 

C4. The finding of IgM being greater in contacts than in active TB is interesting, suggesting the chronicity of active TB drives class-switching. Could a mechanism be included in the discussion?

Response: Thanks for pointing this out. We have modified the Discussion as suggested.

C5. The r-squared and p values should be inserted in the scatter plot figures (correlations).

Response: This has been done.

C6. The results regarding IgM not detectable against Acr is likely technical (concentration of Acr coating?), but the authors have addressed this clearly in the discussion. In the way it is presented, this could still be of benefit for the research community.

(No response required)

C7. Line 268: Can immunoglobulin levels truly be directly compared on an OD basis given it is being compared across different antigens and secondary antibodies? Would this not require a standard curve of subclasses to derive a specific concentration of immunoglobulin? However, it could be correct to state that IgG and then later for figure 6, IgG2 (line 316), is the most readily detectable immunoglobulins in their given assays.

Response: Thanks for pointing this out. The words ‘most abundant’ (line 268) and ‘dominant’ (line 316) have now been replaced by ‘most readily detectable’.

Minor comments:

1. Line 66: What is meant by ‘presumptive’ in single quotation marks?

R: It emphasizes the need for exercising caution while interpreting TST/IGRA results.

2. Line 88: Missing a word – Mtb membrane lysate or proteins?

R: Proteins (correction done).

3. Line 97: Why is ‘Mtb complex’ in single quotation marks? 

R: Quotation marks have been removed.

4. Line 95-96: What is meant by sustain the bacilli in 

R: Some of the dictionary meanings of sustain are: ‘to keep alive’, ‘strengthen’ or ‘support’.

5. Line 188: Should be IgA not AgA

R: Corrected.

6. Line 200: The concentration of the secondary should be included.

R: Included.

7. Line 256: Missing word – elaborated ‘on’.

R (line 276): Replaced the word elaborated with described.

8. Line 394: It is unclear what is meant by quantum of infection increasing.

R: We have replaced the word quantum with burden.

9. Line 556: Typo in semi-quantitative. The point with semi-quantitative is that levels of antibodies cannot be directly compared when using different reagents (secondaries) or assays. Further, it also poses a challenge for defining cut-off values for diagnostic tests as the units are arbitrary and contingent on the individual assay. However, it is accepted that this study is exploratory in nature.

R: Typo in line 566 has been corrected.

10. Line 451: Why is dynamic in single quotation marks?

R: Single quotation marks have been removed.

11. Line 492: Why is ‘cell-surface’ in single quotation marks?

R: To emphasize that ‘cell-surface’ is an ill-defined entity.

12. Line 512: “gets worsened” should perhaps be “is worsened”

R: Corrected.

Reviewer #2: My comments about some of the responses given by authors

Author’s response: We have used TST as the reference assay, reasons for which have now been elaborated more explicitly in revised Introduction (2nd paragraph).

Authors have explained the reason why they used TST as the reference assay. I may consider this a valid reason, but it would be easier to compare their results with others reports using IGRAs. Still the data provided in this paper may be useful to complement the information about this topic.

(No response required)

Author’s response: We fully agree with this observation. In fact we have not proposed that the crude membrane should be used as such. In concluding paragraph of Discussion we had said that “In addition to Acr, the study draws attention to MtM as an emerging source of immunodominant T cell antigens”. We have now revised the Discussion to further clarify this point. However, in real life the ‘non-tuberculous mycobacteria’ as well as BCG do not seem to confound the results of TST which also uses a crude antigen (Farhat et al, Int J Tuberc Lung Dis, 2006; 10:1192).

I agree with the paragraph mentioned in discussion. But the fact that MtM is a rich source of antigens, most of them unstudied, reveals that it is not close to determine which protein may actually function as biomarker. As author also responded: we have made some such attempts in the past (ref 8) but have not been able meet any remarkable success.

(No response required)

Additional comment:

I expected to see the changes made to the paper highlighted to facilitate the second review, instead I had to go through all the paper to see the changes

Response: As required by the journal, we had also submitted a copy of the manuscript showing corrections.

---

## [Editor Report · Decision Letter 2]

14 Jan 2020

Immune responses to Mycobacterium tuberculosis membrane-associated antigens including alpha crystallin can potentially discriminate between latent infection and active tuberculosis disease

PONE-D-19-27417R2

Dear Dr. Sinha,

We are pleased to inform you that your manuscript has been judged scientifically suitable for publication and will be formally accepted for publication once it complies with all outstanding technical requirements.

With kind regards,

Katalin Andrea Wilkinson, PhD

Academic Editor

PLOS ONE
---

## [Editor Report · Acceptance letter]

15 Jan 2020

PONE-D-19-27417R2 

Immune responses to *Mycobacterium tuberculosis* membrane-associated antigens including alpha crystallin can potentially discriminate between latent infection and active tuberculosis disease 

Dear Dr. Sinha:

I am pleased to inform you that your manuscript has been deemed suitable for publication in PLOS ONE. Congratulations! Your manuscript is now with our production department. 

With kind regards,

on behalf of

Associate Professor Katalin Andrea Wilkinson 

Academic Editor

PLOS ONE